# TIME-DEPENDENT MIRROR FLOWS AND WHERE TO FIND THEM

## ABSTRACT

Explicit regularization and implicit bias are often studied separately, though in practice, they act in tandem. However, their interplay remains poorly understood. In this work, we show that explicit regularization modifies the behavior of implicit bias and provides a mechanism to control its strength. By incorporating explicit regularization into the mirror flow framework, we present a general approach to better understand implicit biases and their potential in guiding the design of optimization problems. Our primary theoretical contribution is the characterization of regularizations and reparameterizations that induce a time-dependent Bregman function, with a discussion of the implications of its temporal variation. Importantly, our framework encompasses single-layer attention, and application to sparse coding. Extending beyond our core assumptions, we apply this framework to LoRA finetuning, revealing an implicit bias towards sparsity.

## 1 INTRODUCTION

Regularization is a fundamental technique in machine learning that helps control model complexity, prevent over-fitting and improve generalization (Kukačka et al., 2017). There are various ways to regularize a model (Santos & Papa, 2022), including weight decay, the lasso penalty, dropout, initialization strategies, early stopping, model constraints, and the introduction of noise. In this paper, we focus on two major categories of regularization: explicit regularization and implicit bias, as well as their interaction. We introduce both concepts within a general minimization problem. Consider the objective function $f\colon \mathbb{R}^n \to \mathbb{R}$ to be minimized with respect to $x$:

$$\min_{x\in\mathbb{R}^n} f(x). \tag{1}$$

In the context of explicit regularization, a penalty term $h(x)$ is incorporated into the objective function, directly modifying the learning algorithm to prevent overfitting (Goodfellow et al., 2016), as follows:

$$\min_{x\in\mathbb{R}^n} f(x) + \alpha h(x). \tag{2}$$

This approach constrains the model's complexity and encourages simpler solutions that are more likely to generalize well to new, unseen data (Tian & Zhang, 2022). Common explicit regularization methods include $L_1$ (LASSO) and $L_2$ (Weight decay) regularization (Bishop & Nasrabadi, 2006). The effectiveness of explicit regularization techniques has been demonstrated across various machine learning paradigms (Arpit et al., 2016), including supervised learning, unsupervised learning, and reinforcement learning.

Implicit bias (Gunasekar et al., 2018), can be considered as an inherent aspect of the model design that does not require explicit modifications to the objective function. The goal of characterizing the implicit bias is to understand how overparameterization impacts the training dynamics and, thus, model selection. For example, in the presence of many global minima, optimization algorithms like gradient descent inherently guide the solution toward specific global minima that enjoy some type of low norm property (Pesme et al., 2021).

Consequently, the learned model's properties, including its generalization performance, are significantly influenced by the choice of model. For example, (Stochastic) gradient descent provably converges to the solution with the lowest $L_1$ distance from the initialisation for overparameterized least-squares regression (Pesme et al., 2021).

Such an implicit bias is often associated with a mirror flow (Karimi et al., 2024; Li et al., 2022), which results from a reparameterization of $f$ by setting $x = g(w)$, where $w \in M$ and $M$ is a smooth manifold. Note the fundamental difference explicit regularization in the original space and the mirror flow with the objective function:

$$\min_{w \in M} f(g(w)) + \alpha h(w). \tag{3}$$

The explicit regularizer $h$ acts on the parameters $w$ and not $x = g(w)$. Our main goal is to understand how the explicit regularization $\alpha h(w)$ affects the implicit bias and thus the effective regularization in the original parameter space $x$. While both explicit regularization and the mirror flow framework have been extensively studied independently, the goal of this paper is to analyze their interplay and to show how the explicit regularization affects the implicit bias by integrating explicit regularization into the mirror flow framework. This integration will allow us to gain valuable insights into different problems like sparse coding, attention, and LoRA. Since the nature and strength of implicit bias are usually constant throughout training and inherently determined by the reparameterization, they can sometimes degrade performance or simply not fit to a learning task. As we show, however, they can be adapted and controlled by explicit regularization, which induces a time dependent mirror flow. In previous work, it has been shown that overparametrization leads to lower-rank solutions or $L_1$ bias (Arora et al., 2019; Pesme et al., 2021; Vasudeva et al., 2024), which reveals a bias towards sparsity in particular settings. Nevertheless, factors such as finite learning rates and noise can obscure this sparsity bias. In this work, we demonstrate that explicit regularization offers a mechanism to control this bias effectively. Specifically, it modulates the sparsity bias via the time-dependent mirror flow.

More generally, we aim to integrate explicit regularization into the mirror flow framework, thereby unifying these two concepts. We provide sufficient conditions for the reparameterization $g$ and explicit regularization $h$ similar to (Li et al., 2022), to analyze the resulting optimization problem within the extended mirror flow framework and obtain convergence results. Additionally, we characterize the regularization $h$ in terms of $g$ to understand their interplay and impact on the Legendre function, which can be associated with the implicit bias. Concretely, we identify three distinct effects:

- Type of bias: the explicit regularization changes the shape of the Legendre function. For example, the shape changes from an $L_2$ norm to $L_1$ norm.

- Positional bias: the explicit regularization shifts the global minimum of the Legendre function. For the standard Legendre function, the global minimum corresponds to the network's initialization (Li et al., 2022). During training, the explicit regularization moves the minimum closer to zero.

- Range shrinking: the explicit regularization shrinks the range of the attainable values for the Legendre function. For example, the $L_1$ norm of the network parameters becomes fixed during training.

The effects are illustrated in Figure.1. We further analyze the importance of explicit regularization and its effect on implicit bias through multiple experiments, including sparse coding, attention mechanisms in transformers, and LoRA. In the latter two cases, we observe that large weight decay leads to rank collapse. Additionally, our findings also suggest a strategy to mitigate performance degradation resulting from this collapse. While (Dai et al., 2021) and (Khodak et al., 2022) have studied the effect of constant explicit regularization on the representation cost and quadratic reparameterizations, we are concerned with the effect of both constant or dynamic explicit regularization on the implicit bias. Our contributions are summarized as follows:

- We provide sufficient conditions for incorporating different types of explicit regularization into the mirror flow framework and characterize their effect, focusing on three key impacts on the implicit bias: positional bias shift, type of bias, and range shrinking, which can pose challenges for trainability.

- We propose a systematic procedure for identifying these regularizations and establish a general convergence result for the framework, which suggests how to overcome the above challenges by changing the explicit regularization.

- We highlight the effects of the regularization and the resulting implicit bias in experiments such as sparse coding, attention in transformers, and LoRA fine-tuning in large language models.

- Particularly, we obtain the insight that weight decay controls the sparsification strength induced by quadratic reparameterizations such as attention and LoRA.

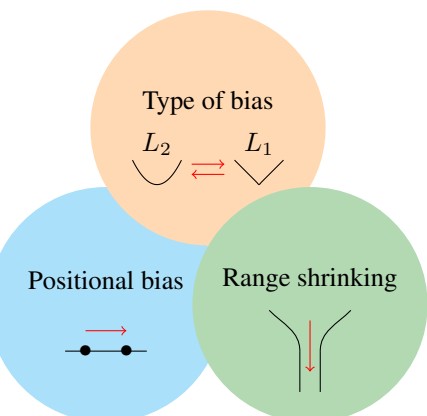

Figure 1: Illustration of three established effects of explicit regularization ($\rightarrow$) on implicit bias.

## 2 RELATED WORK

**Regularization** There are multiple ways of regularizing training in supervised learning. Some of the most widely used techniques include weight decay (Krogh & Hertz, 1991), data augmentation (Cubuk et al., 2020; Orvieto et al., 2023), dropout (Srivastava et al., 2014), and batch normalization (Ioffe & Szegedy, 2015). Weight decay, or $L_2$ regularization, discourages large weights to reduce overfitting. Data augmentation enhances the diversity of training examples by applying random transformations—such as rotations, flips, and crops—to the input data, helping neural networks to generalize better. Dropout randomly deactivates a subset of neurons during each iteration, simulating ensemble learning by creating multiple network configurations. Finally, batch normalization normalizes the inputs to each layer in a mini-batch by subtracting the mean and dividing by the standard deviation, ensuring that inputs are consistently centered and scaled during training.

**Implicit bias** The implicit bias is a well-studied phenomenon (Woodworth et al., 2020; Gunasekar et al., 2017a; 2020; Li et al., 2022), which has primarily been characterized within the mirror flow framework, a well-established concept in convex optimization (Alvarez et al., 2004; Beck & Teboulle, 2003; Rockafellar & Fenchel, 1970; Boyd & Vandenberghe, 2009), which we extend by explicit regularization that can induce a time-dependent Bregman function. A mirror flow can be interpreted as a gradient flow on a Riemannian manifold (Li et al., 2022; Alvarez et al., 2004), which has also been derived for stochastic gradient descent (Pesme et al., 2021; Even et al., 2023). The study of discrete versions (Sun et al., 2022) has led to novel algorithmic designs (Raj & Bach, 2021; González et al., 2024; Azizan et al., 2022). Time-dependent mirror descent, however, is largely underexplored, except for an analysis of some of its intrinsic properties and an application to continuous sparsification (Radhakrishnan et al., 2021; Jacobs & Burkholz, 2024).

**Applications of the mirror flow framework** The mirror flow framework has been applied to various architectures, including attention mechanisms in transformers (Vaswani, 2017; Vasudeva et al., 2024; Sheen et al., 2024), matrix factorization (Li et al., 2021; Gunasekar et al., 2017b; 2020) and diagonal linear networks (Li et al., 2022; Pesme et al., 2021; 2024; Woodworth et al., 2020). For deep matrix factorization, implicit bias has also been studied using gradient flow methods (Marion & Chizat, 2024; Arora et al., 2019). These studies indicate that the flow tends to be implicitly biased toward solutions with lower rank or low nuclear norms. We demonstrate that explicit $L_2$ regularization further enhances its strength, for example in the context of quadratic overparameterization. This is illustrated through experiments on transformer networks. Moreover, we identify the inherent bias of Low-Rank Adaptation (LoRA) (Hu et al., 2021; Wan et al., 2024) and delve into the challenges that are associated with it. This is especially of interest, as LoRA has gained significant popularity in the field of large language models (LLMs) as it allows for cost-effective finetuning.

**Sparse coding** Sparse coding (SC) is a powerful representation technique widely employed in signal processing and pattern recognition (Zhang et al., 2015). It seeks to represent observations as a linear combination of fundamental elements, termed atoms, which collectively form a dictionary.

The core principle of SC is to achieve a sparse representation by imposing constraints, typically using the $L_0$-norm. However, this formulation leads to an NP-hard problem (Tropp, 2004). An alternative strategy relaxes the constraint to the $L_1$-norm, transforming the original problem into a convex, albeit non-smooth optimization task. Proximal algorithms have proven effective to solve these non-smooth problems (Daubechies et al., 2004). Furthermore, convolutional sparse coding (CSC) extends SC by modeling the dictionary as a concatenation of circulant matrices. Notably, CSC has a strong connection with modern convolutional neural networks (CNNs), where the forward pass of a CNN can be viewed as a thresholding pursuit for a multi-layer CSC model (Papyan et al., 2017). This relationship provides valuable insights into the theoretical underpinnings of CNNs and their connection to sparse representation techniques.

## 3   EXPLICIT REGULARIZATION IN THE IMPLICIT BIAS FRAMEWORK

To analyze the impact of regularization on the training dynamics, we first present the gradient flow corresponding to our optimization problem in Eq. (1). The implicit bias is then characterized by the mirror flow framework, which is the stepping stone for our extension.

Consider the optimization problem in Eq. (1). The gradient flow for the training dynamics is:

$$dw_t = -\nabla_w f(g(w_t))dt \qquad w_0 = w_{init},$$

where $\nabla_w$ is the gradient with respect to $w$. For a specific choice of $g$, reparameterizing the loss function $f$ leads to a mirror flow. A general framework is given in (Li et al., 2022) to study the implicit bias through a mirror flow. We provide a summary in Appendix A. Formally, let the reparameterization $g$ be regular (Definition A.1), commuting (Definition A.3) and satisfy Assumption A.1. Then, by Theorem A.1, there is an implicit regularizer $R: \mathbb{R}^n \to \mathbb{R}$ that follows the dynamics:

$$d\nabla_x R(x_t) = -\nabla_x f(x_t)dt, \qquad x_{init} = g(w_{init}). \qquad (4)$$

$R$ is a Legendre function that is associated with the implicit bias in the optimization. For example, $R$ can be the hyperbolic entropy encountered in Pesme et al. (2021); Woodworth et al. (2020). Depending on the initialization of the reparameterization, the entropy is equivalent to either $L_2$ or $L_1$ implicit regularization. The equivalence to $L_1$ is associated with the so-called feature learning regime, which has been argued to improve generalization performance, highlighting a positive impact of overparameterization on deep learning. Notably, by introducing explicit regularization, the Legendre function $R$ can change over time, which has only been encountered by Jacobs & Burkholz (2024) in a specific setting, where it was crucial to exploit the implicit bias for gradual sparsification.

Accordingly, in the reparameterized setting of Eq. (3) with parametrization $g$ and explicit regularization $h$, we allow the regularization parameter $\alpha$ to vary over time during the gradient flow, denoted as $\alpha_t$. This induces the following gradient flow:

$$dw_t = -\left(\nabla_w f(g(w_t)) + \alpha_t \nabla_w h(w_t)\right) \qquad w_0 = w_{init}.$$

To rigorously define the corresponding time-dependent mirror flow, we define a parameterized Legendre function similar to Definition 3.8 (Li et al., 2022).

**Definition 3.1** *Let $A$ be a subset of $\mathbb{R}$. A parameterized Legendre function is $R_a: \mathbb{R}^n \to \mathbb{R}^n$ such that for all $a \in A$, $R_a$ is a Legendre function (Definition 3.8 (Li et al., 2022)).*

Definition 3.1 and Theorem A.1 enable us to state our main result, as follows.

**Theorem 3.1** *Let $g: M \to \mathbb{R}^n$ and $h: M \to \mathbb{R}$ be regular and commuting reparameterizations satisfying Assumption A.1. Then there exists a time-dependent Legendre function $R_a$ such that*

$$d\nabla_x R_{a_t}(x_t) = -\nabla_x f(x_t)dt, \qquad x_0 = g(w_{init}) \qquad (5)$$

*where $a_t = -\int_0^t \alpha_s ds$. Moreover, $R_a$ only depends on the initialization $w_{init}$ and the reparameterization $g$ and $h$, and is independent of the loss function $f$.*

Proof. See Theorem B.1 in the appendix. The main steps of the proof are:

- Applying Theorem 4.9 (Li et al., 2022) to the time-dependent loss function $L_t(x, y) = f(x) + \alpha_t y$ with the reparameterization $x = g(w)$ and explicit regularization $y = h(w)$ to get the resulting mirror flow with Legendre function $R(x, y)$.

- Utilizing that $R$ is strictly convex to show that $y \to \partial_y R(x, y)$ is invertible.

- We use that the mirror flow for $y_t$ is defined by $\partial_y R(x_t, y_t) = a_t$, where $a_t = -\int_0^t \alpha_s ds$. Plugging in the inverse $y_t = Q(x_t, a_t)$, into $\nabla_x R(x_t, y_t)$, to get an expression for the gradient of the time dependent Legendre function. This gives the equation for the time dependent mirror flow $\nabla_x R(x_t, Q(x_t, a_t)) = \mu_t$, where $\mu_t = -\int_0^t \nabla_x f(x_s) ds$.

- In the final step, showing that $\nabla_x R(x, Q(x, a))$, where $\nabla_x$ is the derivative with respect to the first entry, is the gradient of a Legendre function for $a$ fixed.

We examine several key implications of Theorem 3.1. First, we provide a geometric interpretation to offer an intuition of how implicit bias and explicit regularization interact. We then extend the convergence result for mirror flows to time-dependent mirror flows (Theorem 3.2) by introducing the so-called contracting property (see Definition 3.2) to also cover time-varying regularization. If we want to exploit these results and control the implicit bias that is induced by a reparameterization $g$, which is often a given neural network or modeling design choice, we have to choose an appropriate regularizer $h$ and thus face the question: Can we characterize the explicit regularizer $h$, given a reparameterization $g$? To demonstrate the versatility and discuss the limitations of our framework, we explore new parametrizations in Appendix B.1. In the following section, we explore the practical implications of these results on previously studied reparameterizations Woodworth et al. (2020); Pesme et al. (2021); Gunasekar et al. (2017a) and study the three main effects of explicit regularization: positional bias, type of bias, and range shrinking.

**Geometric interpretation** Mirror flow can be interpreted as a gradient flow on a Riemannian manifold (Li et al., 2022; Alvarez et al., 2004). If a Legendre function satisfies a mirror flow, the iterates $x_t$ follow the dynamics:

$$dx_t = -\left(\nabla_x^2 R(x_t)\right)^{-1} \nabla_x f(x_t) dt \qquad x_0 = g(w_{init}). \tag{6}$$

This is as a gradient flow on a Riemannian manifold, where the metric is given by $\left(\nabla_x^2 R\right)^{-1}$. In the same way, Theorem 3.1 leads to a new geometric interpretation for regularization. $x_t$ follows:

$$dx_t = -\left(\nabla_x^2 R_{a_t}(x_t)\right)^{-1}\left(\nabla_x f(x_t) + \alpha_t \nabla_x y_t\right) dt \qquad x_0 = g(w_{init}) \text{ and } y_0 = h(w_{init}), \tag{7}$$

where $y_t$ is defined as in Theorem 3.1. This suggests that regularization can be interpreted as a gradient flow with a changing Riemannian metric and a regularization on the manifold. The unexpected result is that the metric evolves due to the time-dependent Legendre function. In practice, we can steer $a_t$ and can thus control the implicit bias. This creates a novel connection between explicit regularization and implicit bias. Another interpretation of this connection is that the effect of the explicit regularization gets stored in the time-dependent Legendre function. Therefore, the explicit regularization has a lasting effect on the training dynamics when it gets turned off. The geometric interpretation not only provides valuable intuition but helps to show convergence for time-dependent Bregman functions, which we define in Definition 3.2 (Definition A.6 (Li et al., 2022)).

**Definition 3.2** *Let $A$ be a subset of $\mathbb{R}$. A parameterized Bregman function is $R_a : \mathbb{R}^n \to \mathbb{R}^n$ such that for all $a \in A$, $R_a$ is a Bregman function (Definition A.6 (Li et al., 2022)). Furthermore, $R_a$ is called contracting if $\frac{dR_a}{da} \leq 0$ for $a \in A$.*

An example of a function that satisfies Definition 3.2 is $R_a(x) = (x - a)^2$. The function is contracting on the set $A = (-\infty, 0]$.

**Remark 3.1** *Note that if there is a $T > 0$ such that for $t \geq T$, $\alpha_t = 0$. We recover a gradient flow with Riemannian metric $\left(\nabla_x^2 R_{a_T}\right)^{-1}$*

We use Definition 3.2 and Remark 3.1 to show convergence for decaying regularization, i.e., $\alpha_t \to 0$.

**Theorem 3.2** *Consider the same settings as Theorem 3.1. Additionally, assume that for $\alpha_t \geq 0$ there is a $T > 0$ such that for $t \geq T$, $\alpha_t = 0$. Moreover, for $a \in [b, 0]$, $R_a$ is a contracting Bregman function for some $b < 0$. Assume that for all $t \geq 0$ the integral $a_t := -\int_0^t \alpha_s ds \geq b$. For the loss function assume that $\nabla_x f$ is locally Lipschitz and $\text{argmin}\{f(x) : x \in \text{dom} R_{a_\infty}\}$ is*

*non-empty. Then the following holds: if $f$ is quasi-convex, $x_t$ converges to a point $x_*$ which satisfies $\nabla_x f(x_*)^T (x - x_*) \geq 0$ for $x \in domR_{a_\infty}$. Furthermore, if $f$ is convex, $x_*$ converges to a minimizer $f$ in $\overline{domR_{a_\infty}}$.*

For the proof, we refer to Theorem B.2 in the appendix. The proof is split into two parts:

- Showing that the iterates are bounded up to time $T$ with the contracting property and quasi-convexity.
- Demonstrating convergence after time $T$, using the geometric interpretation of evolution of $x_t$.

Theorem 3.2 demonstrates that the contracting property enables us to show convergence for time-dependent Bregman functions. In addition, Theorem 4.17 and Corollary 4.18 in (Li et al., 2022) for diagonal linear networks can be recovered within this framework. These results show that $x_* = \arg\min_{x \in domR_{a_\infty}} R_{a_\infty}$. Therefore, since we control $\alpha_t$, we also control the function that is implicitly minimized.

**Reparameterizations** We characterize the regularization $h$ for commuting and regular reparameterization classes $g$ such that Theorem 3.1 applies. First, we present a result for separable reparameterizations $g$, which encompass all previous settings Woodworth et al. (2020); Pesme et al. (2021); Gunasekar et al. (2017a).

**Corollary 3.1** *Let $g$ be a seperable reparameterization such that $g_i(w_i) = \sum_{j=1}^{m_i} g_{i,j}(w_{i,j})$ and $h(w) = \sum_{i=1}^{n} \sum_{j=1}^{m_i} h_{i,j}(w_{i,j})$, where $g_{i,j} : \mathbb{R} \to \mathbb{R}$ and $h_{i,j} : \mathbb{R} \to \mathbb{R}$. Furthermore, assume that $g$ and $h$ are analytical functions. Then if and only if $h$ and $g$ satisfy*

$$h_{i,j} = c_{i,j} g_{i,j} \qquad \forall i \in [n], j \in [m_i],$$

*with $c_{i,j} \in \mathbb{R}$ a constant, Theorem 3.1 applies.*

Proof. The result follows from the commuting relationship between $g$ and $h$. The Wronskian between two analytical functions is zero if and only if they are linearly dependent (Bôcher, 1901). □

Modern machine learning tends to rely heavily on over-parameterization. Our next primary focus is to demonstrate the advantages of Theorem 3.1 in this common context of over-parameterized parameterizations. We provide examples showcasing the positive applications of Corollary 3.1.

**Example 3.1** *The reparameterization $u^2 - v^2$ with regularizations of the form $c_u u^2 - c_v v^2$. Setting $c_u = 1$ and $c_v = -1$ leads to the $L_2$ regularization on the reparameterization.*

Example 3.1 has been used to study the effect of stochasticity on overparameterized networks Pesme et al. (2021). More generally, we present a general class of examples that always results in a well-posed optimization problem, in this case $h$ is positive.

**Example 3.2** *Consider the reparameterization $a(u) - b(v)$, where $a$ and $b$ are positive analytical increasing functions. In this case, the regularization $c_u a(u) - c_v b(v)$ can always be employed. By selecting $c_u \geq 0$ and $c_v \leq 0$, the optimization problem remains well-posed.*

This approach encompasses reparameterizations such as $u^{2k} - v^{2k}$ (Woodworth et al., 2020) and new $\log u - \log v$. Next, we will discuss another significant class of reparameterizations: the quadratic reparameterizations, as described in Theorem 4.16 in (Li et al., 2022).

**Theorem 3.3** *In the setting of Theorem 3.2, consider the commuting quadratic parametrization $G$: $\mathbb{R}^D \to \mathbb{R}^d$ and $H$: $\mathbb{R}^D \to \mathbb{R}$, where each $G_i(w) = \frac{1}{2}w^T A_i w$ and $H(w) = \frac{1}{2}w^T B w$, for symmetric matrices $A_1, A_2, \ldots, A_d \in \mathbb{R}^{D \times D}$ and symmetric matrix $B \in \mathbb{R}^{D \times D}$ that commute with each other, i.e., $A_i A_j - A_j A_i = 0$ for all $i, j \in [d]$ and $BA_j - A_j B = 0$ for all $j \in [d]$. For any $w_{init} \in \mathbb{R}^D$, if $\nabla_w G_i(w_{init})_{i=1}^{d} = A_i w_{init}_{i=1}^{d}$ and $\nabla_w H(w_{init}) = B w_{init}$ are linearly independent, then the following holds:*

- $Q_a(\mu) = \frac{1}{4}|| \exp(aB + \sum_{i=1}^{d} \mu_i A_i) w_{init}||_{L_2}^2$ *is a time-dependent Legendre function with domain $\mathbb{R}^d$.*

- *For all $a \in \mathbb{R}$, $R_a$ is Bregman function with $dom R_a = \overline{range \nabla_x Q_a}$. Furthermore, if $B$ is positive semi-definite, then $\frac{dR_a}{da} \leq 0$, therefore Theorem 3.2 applies.*

Proof. The first statement is derived by applying Theorem 4.16 from (Li et al., 2022). The second statement follows from recognizing that exp(aB) acts as a linear transformation of the initialization $w_{init}$. Subsequently, applying Theorem 4.16 of (Li et al., 2022) gives the first part of the last statement. It remains to demonstrate that $R_a$ is contracting. Since $B$ is positive semi-definite, it follows that $\frac{d}{da}Q_a \geq 0$. By the reverse ordering property of convex conjugation, we have that $\frac{d}{da}R_a \leq 0$. For completeness, let $h > 0$; then for $a \in \mathbb{R}$, we have $Q_{a+h} \geq Q_a$. Applying the reverse ordering property implies $R_{a+h} \leq R_a$. Rearranging and dividing by $h$ gives $\frac{1}{h}(R_{a+h} - R_a) \geq 0$. Taking the limit as $h \to 0$ concludes the proof. $\square$

Theorem 3.3 encompasses recent works on the reparameterization $m \odot w$, which has been proposed to sparsify neural networks (Jacobs & Burkholz, 2024), and extends work on analyzing transformers (Vasudeva et al., 2024). Note, the operation $\odot$ is pointwise multiplication (Hadamard product). Having identified classes where we can determine $h$, we will now apply our results to illustrate how time dependence influences the dynamics and gain novel insights.

**Remark 3.2** *It is important to note that for the time-dependent Bregman function in Theorem 3.3 to be contracting, $B$ needs to be positive semi-definite. Furthermore, $B = I$ corresponds to $L_2$ regularization on the reparameterization. We will utilize this to examine the dynamics for the key and query matrices $K$ and $Q$ in vision transformer networks with $L_2$ regularization and LoRA.*

## 4 THE EFFECT OF REGULARIZATION

In this section, we introduce several time-dependent Legendre functions to demonstrate the wide applicability of our analysis. We aim to gain insights into how regularization affects implicit bias, focusing on two primary effects: the alteration of positional bias and the type of regularization. For instance, changing the type of regularization corresponds to transitioning from implicit $L_2$ to $L_1$ regularization. Furthermore, we illustrate a third effect, which examines how regularization shrinks the range of mirror flow. We consider the reparameterizations $m \odot w$ and $u^{2k} - v^{2k}$ for $k \in \mathbb{N}^+$. Using $L_2$ regularization, represented as $||m||_{L_2}^2 + ||w||_{L_2}^2$, is permissible according to Theorem 3.3 for $m \odot w$. According to Corollary 3.1, we can also apply the regularization $\sum_{i=1}^n u_i^{2k} + v_i^{2k}$. The parameterization $m \odot w$ illustrates the effect of $L_2$ regularization on attention mechanisms, while $u^{2k} - v^{2k}$ highlights the range shrinking. For the parameterizations $m \odot w$, we provide the time-dependent Legendre function explicitly. In contrast, an analytic expression for $u^{2k} - v^{2k}$ is not feasible (Woodworth et al., 2020). However, we can still examine the evolution of the gradient flow. In addition we found a new parametrization that has $L_1$ to $L_2$ type of bias change, see Appendix B.1.

**The parameterization $m \odot w$** Consider the separable parameterization $x = m \odot w$ with regularization $y = ||m||_{L_2}^2 + ||w||_{L_2}^2$ as discussed in (Jacobs & Burkholz, 2024). For initializations where $|w_{init}| < m_{init}$, Theorem 3.3 holds. We can compute the time-dependent Bregman function:

$$R_a(x) = \frac{1}{4} \sum_{i=1}^d x_i \text{arcsinh}\left(\frac{x_i}{2\exp(2a)u_{i,0}v_{i,0}}\right) - \sqrt{x_i^2 + 4\exp(4a)u_{i,0}^2 v_{i,0}^2} - x_i \log\left(\frac{u_{i,0}}{v_{i,0}}\right) \quad (8)$$

where $u_0 = (m_{init} + w_{init})/\sqrt{2}$ and $v_0 = (m_{init} - w_{init})/\sqrt{2}$. We recover the corrected hyperbolic entropy (Woodworth et al., 2020), which now is dependent on $a$. Note, that we used Theorem 3.3 to find $R_a$, we can invert the corresponding function $Q_a(\mu)$, where $\mu = -\int_0^t \nabla_x f(x_s) ds$. The regularization thus affects the time dependent Legendre function trough changing $a$. This allows us to modulate between an implicit $L_2$ and $L_1$ regularization through explicit regularization (Jacobs & Burkholz, 2024). Moreover, $a$ also controls the location of the global minimum, a smaller $a$ corresponds to moving it closer to zero. Therefore, we both change the type of bias and positional bias. In Appendix C Figure.9a we illustrate the type of bias and positional bias effects for $m \odot w$.

Building on this result, we study attention, where the query $Q$ and key $K$ matrices are both of size $n \times n$. Consequently, the matrix $K^T Q$ is a quadratic parameterization. We use $L_2$ regularization, ($B = I$) and assume that the corresponding matrices $A_j$ for $j \in [n^2]$ satisfy the conditions of

Theorem 3.3, Specific choices for these matrices will be illustrated later. It is worth noting that a transformer also has a value matrix $V$ and an activation function. Assuming $V$ is not trainable and that the function $f$ encompasses the activation function, the gradient flow dynamics are described by Theorem 3.1, which characterizes the implicit bias. For optimality and convergence, however, additional assumptions must be satisfied (Sheen et al., 2024). Specifically, consider the optimization problem for a transformer network with $L_2$ regularization:

$$\min_{K,Q} f(K^T Q) + \alpha \left( ||K||^2_{fro} + ||Q||^2_{fro} \right).$$

When both $K$ and $Q$ are diagonal matrices with values $\Lambda_k$ and $\Lambda_q$, this corresponds to the setting of $m \odot w$. By slight abuse of notation $m = \Lambda_k$ and $w = \Lambda_q$. Thus, the time-dependent Legendre function in Eq. (8) applies. Therefore, the implicit bias of spectrum $\Lambda_k^T \Lambda_q$, is described by Eq. (5). This implies that the $L_2$ regularization modulates the implicit bias of the spectrum between $L_2$ an $L_1$, which corresponds to the Frobenius norm and nuclear norm of the matrix $K^T Q$. This can be generalized by using the so-called alignment property (Sheen et al., 2024). In our experiments, we illustrate that a larger $L_2$ regularization leads to a faster modulation, i.e. minimizing the nuclear norm of $K^T Q$ over the Frobenius norm. Note that $L_2$ regularization is used for transformers to keep the parameters from growing too large. As observed in Khodak et al. (2022), weight decay encourages minimization of the nuclear norm. Nevertheless, this is not the full picture, the weight decay changes the geometry according to Eq. (6), leading to a modulation between the Frobenius and nuclear norm. This reveals how transformers enter the feature regime or become potentially too sparse.

**The parametrization** $u^{2k} - v^{2k}$  In the previous paragraph, we have illustrated with the time-dependent Bregman function of $m \odot w$ that both the type of bias and positional bias can change. In this paragraph, we illustrate another phenomenon. We show that the range of the time-dependent Legendre function can shrink due to the regularization. We consider the parameterization $g(w) = u^{2k} - v^{2k}$ with regularization $h(w) = \sum_{i=1}^{n} u_i^{2k} + v_i^{2k}$ as allowed by Corollary 3.1. Moreover, similar to the previous parameterization, the current parameterization also exhibits both an $L_2$ to $L_1$ type change see Theorem 3 in (Woodworth et al., 2020). Unfortunately, there is no analytical formula available for the Legendre function in this case. Therefore we compute the flow and derive the domain which is the range of the time-dependent mirror flow. The flow $Q_a$ is given by

$$Q_{a_t}(x_t) = \left( (2k-2)(2k) \frac{1}{\mu_t + a_t + c_u} \right)^{\frac{2k}{2k-2}} - \left( (2k-2)(2k) \frac{1}{-\mu_t + a_t + c_v} \right)^{\frac{2k}{2k-2}} \tag{9}$$

where $\mu_t = -\int_0^t \nabla f(x_s) ds$ and $a_t = -\int_0^t \alpha_s ds$. The domain of $Q_a$ is the range of $R_a$. The domain of $Q_a$ depends on $a$ as follows $\mu \in (-c_u - a, c_v + a)$. Since $a_t$ is negative the domain of $Q_a$ is shrinking over time. Thus the range of $R_a$ is also shrinking. By shrinking the range we start excluding the set of acceptable solutions of the original optimization problem minimizing $f$. This may eventually lead to not being able to solve the main optimization problem. Although we have no analytical expression for $R_a$, we can use $Q_a$ to approximate it. We illustrate the range shrinking effect in Appendix C Figure.9b with this approach.

**Take away**  We have illustrated the three effects summarized in Figure.1. The insight we get for both parameterizations is that the positional bias is getting closer to the origin. Furthermore, the implicit bias changes from $L_2$ to $L_1$, entering the "rich regime". Moreover, the range of the implicit bias is shrinking in the case of $u^{2k} - v^{2k}$ for $k > 1$.

## 5 EXPERIMENTS

We highlight three benefits of our theoretical analysis. The first experiment concerns sparse coding, where we show that our results hold for finite learning rate and observe the range shrinking effect. The second experiment focuses on attention in transformers, studying the effect of moving the implicit bias from the Frobenius norm to the nuclear norm. The third experiment is finetuning an LLM with LoRA, where we illustrate the storage of the explicit regularization in the time-dependent Legendre function by turning off the weight decay, which is a novel insight. This illustrates that our insights also extend to settings where our assumptions are not met exactly. Note that the change in positional bias is present in all settings. Moreover, the sparse coding experiment is also repeated for the parameterization that has type of bias change from $L_1$ to $L_2$ in Appendix B.1. Furthermore, we provide a detailed exposition on diagonal linear networks there as well.

**Sparse Coding** To demonstrate the applicability of our analysis, we extend our study to the traditional sparse coding problem, which is commonly solved using proximal gradient methods. Our approach is based on the online dictionary learning algorithm (Mairal et al., 2009), but with our proposed parameterization substituting the standard sparse coding step. For this experiment, we use the Olivetti faces dataset. We denote the dictionary with $D$, labels with $z$, the code with $g(w)$ and regularization with $h(w)$. The feature dimension of $D$ is $n$. The optimization problem is given by

$$\min_w \frac{1}{2n}\|z - Dg(w)\|^2 + \alpha h(w). \tag{10}$$

We use gradient descent to solve the optimization problem in Eq. 10 with learning rate $\eta > 0$.

**The parameterization** $u^{2k} - v^{2k}$ In this context, we parameterize the sparse code as $g(w) = u^{2k} - v^{2k}$ and set the regularization $h(w) = \sum_{i=1}^n u_i^{2k} + v_i^{2k}$ as discussed in the previous section. The parameters are initialized as $u_0 = \frac{1}{2}(\sqrt{x^2 + \beta^2} + x)^{\frac{1}{2k}}$ and $v_0 = \frac{1}{2}(\sqrt{x^2 + \beta^2} - x)^{\frac{1}{2k}}$, where $\beta = 1$, $x \sim \mathcal{N}(0, I_n)$, and all operations are pointwise. We set regularization strength to $\alpha = 0.001$. We explore various values of $n \in [7]$. Throughout the training process, we track two key metrics: the reconstruction mean squared error (MSE) and the nuclear norm of the sparse code, $g(w) = u^{2k} - v^{2k}$. The results are presented in Figure.2. We observe the effect of the range shrinking for $k > 1$, for larger $k$ the evolution of the nuclear norm becomes stationary faster. This indicates that the range in which the time-dependent Legendre function is allowed to move has shrunk. The shrinking also causes the MSE to converge faster for large $k$.

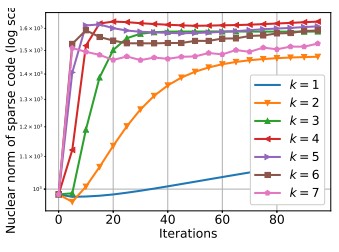 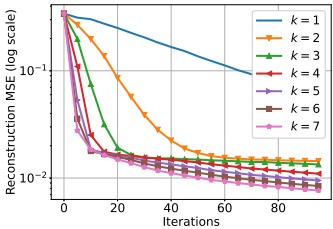

(a) nuclear norm of sparse code $w$             (b) Reconstruction MSE of $x$

Figure 2: Results for sparse coding reparameterisation $g(w) = u^{2k} - v^{2k}$

**Attention in transformers** We leverage the insight from Theorem 3.3 to strengthen the results of (Sheen et al., 2024) on transformers. Our experiments are based on a Tiny-ViT transformer network trained on CIFAR10, applying weight decay $\alpha \in \{0.5, 0.1, 0.05, 0.01, 0.005, 0.001, 0.0005, 0.0001\}$. For each layer, we calculate the nuclear norm and Frobenius norm of the product of the query and key matrices, denoted as $|K^T Q|_{frob}$, $|K^T Q|_{nuc}$, average them across all layers, and compute their ratio, which is visualized in Figure.3a. We observe that the decay of the ratio is associated with increasing weight decay, illustrating the type of bias effect. For small weight decay other factors take precedence and the ratio starts increasing at the end of training, as mentioned in the introduction. In contrast, for larger weight decay we do no see this happen, therefore effectively controlling the ratio. Moreover, Figure.3b suggests that large weight decay can lead to lower validation error. Nevertheless, too large weight decay leads to higher validation loss, which is accompanied with a smaller ratio.

**LoRA** Our following LoRA experiments demonstrate that the insights of Theorem 3.3 extend beyond the specific assumptions of the theorem. We finetune, GPT2 (Radford et al., 2019) with LoRA on the *tiny_shakespeare* (Karpathy, 2015) dataset and train for 500 iterations with two different type of schedules. The first employs a constant weight decay throughout the training process, while the second disables weight decay after 200 iterations. Figure.4 presents the results. We observe in Figure 4a that increasing the weight decay leads to a decay in the ratio, illustrating the change in type of bias. Moreover, when the weight decay is turned off, we see that the ratio still decreases. In contrast, this would not be the case for linear paramterizations, which have an implicit $L_2$ bias. To add to this, the ratio of weight decay 1.0 with turning off intersects with the ratio of 0.5 with constant weight decay only after iteration 400. A similar intersection occurs for 0.2 with turning off and 0.1 with constant weight decay. At the 400th iteration, the cumulative amount of applied weight decay is equal. This intersection after the 400 iteration and the fact that the ratios are non-increasing

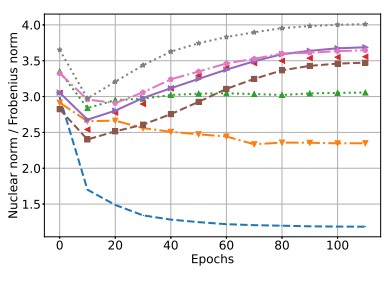
(a) Average ratio $|K^T Q|_{nuc}/|K^T Q|_{frob}$;

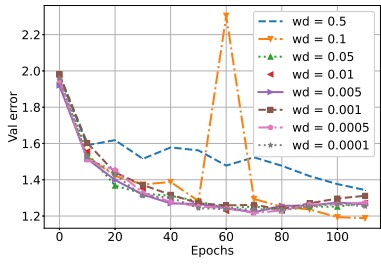
(b) Validation error.

Figure 3: Average ratio and validation accuracy for attention mechanism.

after turning off weight decay illustrate that the regularization is stored within the time-dependent Legendre function. This storage mechanism enables exploration of solutions with lower ratios that are unconstrained by explicit regularization, potentially achieving lower test loss (as shown in Figure 4b). These insights suggest that optimizing dynamic weight decay schedules can lead to improved LoRA fine-tuning outcomes.

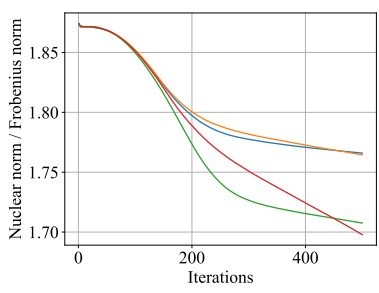
(a) Average ratio $|B^T A|_{nuc}/|B^T A|_{frob}$.

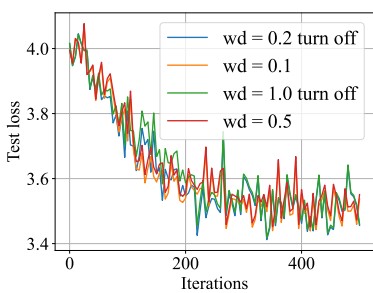
(b) Test loss.

Figure 4: Average ratio and test accuracy for LoRA.

# 6 DISCUSSION

We have provided a framework to analyze how explicit regularization affects implicit bias by integrating it into the mirror flow framework, which has led to a novel geometric interpretation of the interplay between explicit regularization and implicit bias. We have identified sufficient conditions for incorporating these regularizations and characterized their effects on the dynamics, notably positional bias, type of bias, and range shrinking. Additionally, we have established a systematic procedure for identifying suitable regularizations for given parameterizations and established convergence within our framework. We have also illustrated the implications of our theory in the context of sparse coding, attention in transformers, and LoRA fine-tuning. We found that the type of bias can change dynamically during training, for example, from $L_2$ to $L_1$, as observed in our experiments. Accordingly, the geometry of the training dynamics changes as described in Eq. (7). This is associated with a time-dependent Legendre function, which might be of independent interest conceptually. Our findings could have implications also for other regularization methods such as early stopping or for explaining scaling laws that relate the amount of overparameterization to optimal training times. Furthermore, multiple experiments highlight the potential of our framework to not only enhance our understanding of the interplay between explicit regularization and implicit bias but also to pave the way for developing more effective regularization techniques, such as dynamic weight decay, tailored to various model architectures and tasks.

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

## A    IMPLICIT BIAS FRAMEWORK

In this section for completeness we present the existing results for the mirror flow framework. Consider the optimization problem in Eq. (1) for a loss function $f : \mathbb{R}^n \to \mathbb{R}$

$$\min_{x \in \mathbb{R}^n} f(x).$$

We can use the implicit bias framework to study the effect of overparameterization. An overparameterization can be accomplished by introducing a function $g : M \to \mathbb{R}^n$, with $M$ a smooth manifold. For

particular $g$, the reparameterization of the loss function $f$ leads to a mirror flow. A general framework is given in (Li et al., 2022) to study the implicit bias in terms of a mirror flow. Let $R : \mathbb{R}^n \to \mathbb{R}$ be a Legendre function (Definition 3.8 (Li et al., 2022)), then the mirror flow is described by

$$d\nabla_x R(x_t) = -\nabla_x f(x_t)dt, \qquad x_{init} = g(w_{init}) \tag{11}$$

(Li et al., 2022) provide a sufficient condition for the parameterization $g$ such that it induces a mirror flow Eq. (11). The Legendre function $R$ controls the implicit bias.

For this we have to give two definitions that are used to give these sufficient conditions. Furthermore, we define $\partial g$ as the Jacobian of the function $g$. The parameterization has to be regular and commuting we now give the definitions of both these properties.

**Definition A.1** *(Regular Parmeterization Definition 3.4 (Li et al., 2022)) Let $M$ be a smooth submanifold of $\mathbb{R}^D$. A regular parametrization $g : M \to \mathbb{R}^n$ is a $C^1$ parametrization such that $\partial G(w)$ is of rank $n$ for all $w \in M$.*

For the second definition we first need to define what a Lie bracket is.

**Definition A.2** *(Lie bracket Definition 3.4 (Li et al., 2022)) Let $M$ be a smooth submanifold of $\mathbb{R}^D$. Given two $C^1$ vector fields $X, Y$ on $M$, we define the Lie Bracket of $X$ and $Y$ as $[X, Y](w) := \partial Y(w)X(w) - \partial X(w)Y(w)$.*

**Definition A.3** *(Commuting Parameterization Definition 4.1 (Li et al., 2022)) Let $M$ be a smooth submanifold of $\mathbb{R}^D$. A $C^2$ parameterization $g : M \to \mathbb{R}^d$ is commuting in a subset $S \subset M$ iff for any $i, j \in [n]$, the Lie bracket $\left[\nabla g_i, \nabla g_j\right](w) = 0$ for all $w \in S$. Moreover, we call $g$ a commuting parameterization if it is commuting in the entire $M$.*

Besides these two definitions we need to make an additional assumption on the flow of the solution. We define the solution of the gradient (descent) flow of a function $f : M \to \mathbb{R}^n$ initialized at $x \in M$

$$dx_t = -\nabla_x f(x_t)dt \qquad x_0 = x \tag{12}$$

as $x_t = \phi_x^t(x)$ which is well defined if the solution exists. Using this we can make the following assumption.

**Assumption A.1** *(Assumption 3.5 (Li et al., 2022)) Let $M$ be a smooth submanifold of $\mathbb{R}^D$ and $g : M \to \mathbb{R}^n$ be a parameterization. We assume that for any $w \in M$ and $i \in [n]$, $\phi_{g_i}^t(w)$ is well-defined for $t \in (T_-, T_+)$ such that either $\lim_{t \to T_+} ||\phi_{g_i}^t(w)||_{L_2} = \infty$ or $T_+ = \infty$ and similarly for $T_-$. Also, we assume that for any $w \in M$ and $i, j \in [n]$, it holds that for $(t, s) \in \mathbb{R}^2$ that $\phi_{g_i}^s \circ \phi_{g_j}^t(w)$ is well-defined iff $\phi_{g_j}^t \circ \phi_{g_i}^s(w)$*

Using these definitions we state the known result for mirror flow.

**Theorem A.1** *(Theorem 4.9 (Li et al., 2022)) Let $M$ be a smooth submanifold of $\mathbb{R}^D$ and $g : M \to \mathbb{R}^n$ be a commuting and regular parameterization satisfying Assumption A.1. For any initalization $w_{init} \in M$, consider the gradient flow for any time-dependend loss function $L_t : \mathbb{R}^d \to \mathbb{R}$:*

$$dw_t = -\nabla_w L_t(g(w_t))dt, \qquad w_0 = w_{init}.$$

*Define $x_t = g(w_t)$ for all $t \geq 0$, then the dynamics of $x_t$ is a mirror flow with respect to the Legendre function $R$ given by Lemma 4.8 in (Li et al., 2022), i.e.,*

$$d\nabla_x R(x_t) = -\nabla_x L_t(x_t)dt, \qquad x_0 = g(w_{init}).$$

*Moreover, this $R$ only depends on the initialization $w_{init}$ and the parameterization $g$, and is independent of the loss function $L_t$.*

We have used Theorem A.1 to show the Theorem 3.1 in the main text.

## B  PROOFS OF SECTION 3

**Theorem B.1** *Let $g : M \to \mathbb{R}^n$ and $h : M \to \mathbb{R}$ be regular and commuting parameterizations satisfying Assumption A.1. Then there exists a time-dependent Legendre function $R_a$ such that*

$$d\nabla_x R_{a_t}(x_t) = -\nabla_x f(x_t)dt, \qquad x_0 = g(w_{init})$$

*where $a_t = -\int_0^t \alpha_s ds$. Moreover, $R_a$ only depends on the initialization $w_{init}$ and the parameterization $g$ and $h$, and is independent of the loss function $f$.*

Proof. Consider the time dependent loss function $L_t(x, y) = f(x) + \alpha_t y$. Applying Theorem A.1 implies there is a Legendre function $R(x, y)$ such that

$$\begin{cases} \nabla_x R(x_t, y_t) = -\int_0^t \nabla_x f(x_s) ds \\ \partial_y R(x_t, y_t) = -\int_0^t \alpha_s ds. \end{cases} \tag{13}$$

We use Eq. (13) to derive the time dependent Legendre function. First note that $\partial_y \partial_y R(x, y) > 0$ for $(x, y) \in dom R$ since $R$ is strictly convex. This implies that the map $y \to \partial_y R(x, y)$ is invertible. Let the inverse be denoted by $Q(x, a)$, where in the dynamics $a_t = -\int_0^t \alpha_s ds$. Plugging $Q$ into the first part of Eq. (13) gives us

$$\nabla_x R\left(x_t, Q\left(x_t, a_t\right)\right) = -\int_0^t \nabla_x f\left(x_s\right) ds, \tag{14}$$

where $\nabla_x$ is still the derivative with respect to the first entry. Eq. (14) looks already like a time dependent mirror flow. We show now that there exists a Legendre function $R_\alpha$ with the map $\nabla_x R\left(x, Q\left(x, \alpha\right)\right)$ as the gradient. This we can do by showing that the Hessian is symmetric and positive definite and that the $R_\alpha$ is essentially smooth.

By implicitly differentiating, we make the following observation:

$$\frac{dQ}{dx} = -\frac{1}{\partial_y \partial_y R(x, Q)} \nabla_x \partial_y R(x, Q).$$

Next we compute the Hessian and apply observation B:

$$\nabla_x^2 R_\alpha = \nabla_x^2 R(x, Q) + \partial_y \nabla_x R(x, Q) \cdot \frac{dQ}{dx}$$

$$= \nabla_x^2 R(x, Q) - \frac{1}{\partial_y \partial_y R(x, Q)} \partial_y \nabla_x R(x, Q) \nabla_x \partial_y R(x, Q)^T.$$

Observe that this matrix is symmetric as it is a sum of symmetric matrices. It remains to be shown that the Hessian matrix is positive definite. For this we use that $\nabla_x^2 R$ is positive definite. $\nabla_x^2 R$ is PD implies that the inverse $(\nabla_x^2 R)^{-1}$ is PD. The first block entry of this matrix is given by

$$\left(\nabla_x^2 R(x, y) - \frac{1}{\partial_y \partial_y R(x, y)} \partial_y \nabla_x R(x, y) \nabla_x \partial_y R(x, y)^T\right)^{-1}$$

which is also PD. Now this implies the result as the inverse of $\nabla_x^2 R_\alpha$ is PD. It follows that there exists a function $R_a$ such that $\nabla R_a = \nabla_x R(x, Q(x, a))$ by Corollary 16.27 in (Lee, 2013), concluding the first part.

Finally $R_a$ is essentially smooth by construction, using that $R$ is essentially smooth. The boundary $bn(R_a)$ by construction is the set of points $x^*$ that have a sequence $x_n \in domint\nabla_x R(\cdot, Q(\cdot, a))$ such that if $x_n \to x^*$ we have $|||\nabla R|| \to \infty$. Suppose that $R_a$ is not essentially smooth then there exists a sequence $\{x_n\}$ with $x_n \to bd(R_a)$ as $n \to \infty$ such that $\lim_{n\to\infty} ||\nabla_x R(x_n, Q(x_n, a))||^2_{L_2} < \infty$. Nevertheless, $R$ is essentially smooth this implies that

$$\lim_{n\to\infty} ||\nabla_y R(x_n, Q(x_n, a))||^2 = a^2 = \infty,$$

leading to a contradiction. Hence, $R_a$ is a Legendre function with the domain similarly constructed as the boundary. $\square$

**Theorem B.2** *Assume the same settings as Theorem 3.1. Furthermore assume that for $\alpha_t \geq 0$ there is a $T > 0$ such that for $t \geq T$, $\alpha_t = 0$. Moreover for $a \in [b, 0]$, $R_a$ is a contracting Bregman function for some $b < 0$. Assume that for all $t \geq 0$ the integral $a_t := -\int_0^t \alpha_s ds \geq b$. For the loss function assume that $\nabla_x f$ is locally Lipschitz and $argmin\{f(x) : x \in domR_{a_\infty}\}$. Then if $f$ is quasi-convex $x_t$ converges to a point $x_*$ which satisfies $\nabla_x f(x_*)^T (x - x_*) \geq 0$ for $x \in domR_{a_\infty}$. In addition if $f$ is convex $x_*$ converges to a minimizer $f$ in $\overline{domR_{a_\infty}}$.*

Proof. We can bound the trajectory of $x_t$ by using the time dependent Bregman divergence. The divergence between a critical point $x^*$ of $f$ and the itterates $x_t$ is given by

$$D_{a_t}(x^*, x_t) := R_{a_t}(x^*) - R_{a_t}(x_t) - \nabla_x R_{a_t}^T(x^* - x_t) \geq 0$$

Note that the contracting property implies that for $a_2 \leq a_1$ we have $domR_{a_2} \subset domR_{a_1}$. Thus, a critical point $x^*$ in $domR_{a_\infty}$ is in $domR_{a_t}$. Hence, the divergence is well defined. Due to that $f$ is quasi convex and $R_a$ contracting we have that $D_{a_t}(x^*, x_t)$ is bounded. From the contracting property it follows that $R_{a_\infty}(x^*) \geq R_{a_t}(x^*)$. By definition of a Bregman function we have that $x_t$ stays bounded for all $t \geq 0$. It follows that $x_T$ is in the domain of $R_{a_\infty}$ and bounded. Therefore, we have that $D_{a_t}(x^*, x_t) \leq R_{a_\infty}(x^*) - R_{a_t}(x_t) - \nabla_x R_{a_t}^T(x^* - x_t) =: W_t$. Now we show that the evolution of $W_t$ is decaying, implying that $D_{a_t}(x^*, x_t)$ is bounded. The evolution is given by

$$dW_t = \alpha_t \frac{d}{da_t} R_{a_t}(x_t)dt - \nabla_x R_{a_t}(x_t)dx_t + \nabla_x R_{a_t}(x_t)dx_t - d\nabla_x R_{a_t}^T(x^* - x_t)$$
$$\leq +d\nabla_x f(x_t)^T(x^* - x_t)$$
$$\leq 0$$

where we used that $\alpha_t \geq 0$ and the contracting property for the first inequality and quasi-convexity for the second. Therefore $x_t$ stays bounded for $t \in [0, T]$. Now, using the geometeric interpretation Eq. (7) we have that the evolution of $\tilde{x}_t = x_{T+t}$ is a gradient flow on a Riemannian manifold with metric $\left(\nabla_x^2 R_{a_\infty}\right)^{-1}$. Therefore Theorem 4.14 in (Li et al., 2022) applies, which concludes the result. □

### B.1 OTHER PARAMETERIZATIONS

In this section we explore several parameterizations and limitations of the framework. We show that Theorem 3.1 does not apply to linear parametrization. Moreover, Theorem 3.1 does not apply to overparameterizations with depth larger than 2 and weight decay. Nevertheless, we show in experiments that similar effects can occur. We illustrate both the type change and range shrinking effect. Finally, we explore a novel parametrization $\log(u) - \log(v)$. This is to illustrate that the type of bias can also change from $L_1$ to $L_2$.

**Linear parametrization** From Corollary 3.1, we derive another corollary for non-overparameterized parametrization.

**Corollary B.1** *Let $g(x) = x$ be the identity parametrization and $h \in C^2(\mathbb{R}^n, \mathbb{R})$. Then Theorem 3.1 applies if and only if, $h$ is given by $h(x) = \sum_{i=1}^n c_i x_i + d$ where $c_i, d \in \mathbb{R}$ are arbitrary coefficients.*

Proof. To apply the theorem, $h$ needs to be commuting with $g$, implying that $\partial_i \partial_i h = 0 \ \forall i \in [n]$, concluding the result. □

Corollary B.1 poses a limitation in the applicability of Theorem 3.1. Since $h$ is not positive for all $x \in \mathbb{R}^n$, the resulting optimization problem is ill-posed. Therefore, standard non-reparameterized loss functions cannot be analyzed in this manner.

**Beyond quadratic parametrization** We show that the current framework excludes higher order parameterization with weight decay. In order to show that

**Theorem B.3** *Let $g : \mathbb{R}^k \to \mathbb{R}$ be given by $g(w) := \Pi_{i=1}^k w_i$, a $k > 2$ depth reparamterization. Moreover, let $h : \mathbb{R}^k \to \mathbb{R}$ and $h(w) = \sum_{i=1}^k w_i^2$. Then $g$ and $h$ do not commute.*

Proof. This follows directly from checking the commuting condition between $g$ and $h$:

$$[\nabla_w g, \nabla_w h](w) = \nabla_w g(w)\nabla_w^2 h(w) - \nabla_w h(w)\nabla_w^2 g(w)$$

$$= \begin{bmatrix} (4-2k)\Pi_{i\in[k]\setminus\{1\}} w_i \\ \vdots \\ (4-2k)\Pi_{i\in[k]\setminus\{k\}} w_i \end{bmatrix}.$$

In order for this to be equal to zero all products need to be zero. This implies that the gradient flow given by

$$dw_t = - \begin{bmatrix} \Pi_{i\in[k]\setminus\{1\}} w_{i,t} \\ \vdots \\ \Pi_{i\in[k]\setminus\{k\}} w_{i,t} \end{bmatrix} \odot \nabla_x f(g(w_t)) - \alpha_t w_t dt,$$

becomes $dw_t = -\alpha_t w_t dt$ and is independent of $f$. Hence, $g$ and $h$ do not commute $\square$

Theorem B.3 implies that we can not apply Theorem 3.1. We note that the commuting condition is only a sufficient criteria such that a pair $(g, h)$ is a time-dependent mirror flow.

**Experiment over-parameterization** Although, our theoretical result does not hold for parametrization with higher depth we illustrate that the expected effects do occur as well for higher depth. We consider the reparamterization $m \odot w \odot v$ for diagonal linear networks and compare with $m \odot w$, both with weight decay. Moreover, we compare with the parameterization $m$ with $L_1$ regularization to motivate the importance of the geometry, which is controlled by the time-dependent Legendre function. Note, in this setting for $m \odot w$ we can reach the groundtruth (Jacobs & Burkholz, 2024).

Let $d = 40$ be the amount data points and $n = 100$ the dimension of the data. We generate independent data $Z_k \sim N(0, I_n)$ for $k \in [d]$. We assume a sparse ground truth $x^*$ such that $||x^*||_{L_0} = 5$. The training labels are generated by $y_k = Z_k^T x^*$. Moreover, the mean squared error loss function is used. The learning rate $\eta = 10^{-3}$ and we use weight decay $\alpha \in \{0.01, 0.1, 1\}$. We run the 100000 steps with weight decay, after that we run the same amount of steps without weight decay. We initialize $m = \mathbf{0}$ and $w = z = \mathbf{1}$, this ensures that both parametrization are initialized at zero and have the same scaling. In this setup, we illustrate the type change similar predicted for the parametrization $m \odot w$. Moreover, we illustrate the range shrinking which occurs for higher depth parametrization $u^{2k} - v^{2k}$. Note that the ground truth has the following ratio between the $L_1$ and $L_2$ norm 2.23.

In Figure.5a we observe for $m$ that higher weight decay does not get closer to the ground truth after turning the $L_1$ regularization off. This is in line with the fact that the regularization is not stored in the geometry as described by Eq. (7). By turning off the regularization we converge to the closest solution in $L_2$ norm. This is best seen in Figure.6a, where the ratio increases above the value of the ground truth.

In Figure.5b we observe for $m \odot w$ that higher weight decay gets closer to the ground truth after turning the weight decay off. This is in line with the fact that the regularization is stored in the geometry as described by Eq. (7) and a type of bias change from $L_2$ to $L_1$. Furthermore, this is also confirmed in Figure.6b that for large weight decay the ratio gets close to the ratio of the ground truth only after turning the weight decay off. This also illustrates Theorem 3.2.

In Figure.5c, we observe for the regularization strength $1e-1$ a similar effect corresponding to the type of bias change from $L_2$ to $L_1$. In contrast, the higher regularization does not exhibit the same behavior. We claim this is due to the range shrinking effect. To motivate this is not due to the dynamics getting stuck at $x = \mathbf{0}$ we report the final value of first parameter. The value is equal to 1.58 which is not equal to either 0 or the ground truth value 1. To add to this, in Figure.6c we unveil that the ratio for large weight decay stays constant.

In conclusion, the type of bias can improve generalization, whereas $m \odot w$ even goes to the ground truth with high regularization, $m$ does not. Moreover, when we use higher order parametrization such as $m \odot w \odot z$ we encounter a different phenomena: range shrinking. To add to this, higher order parametrization still exhibit the type of bias change in a certain range of regularization strength. Thus, our theoretical framework leads to verifiable predictions. These can be used to improve the training dynamics of neural networks in general.

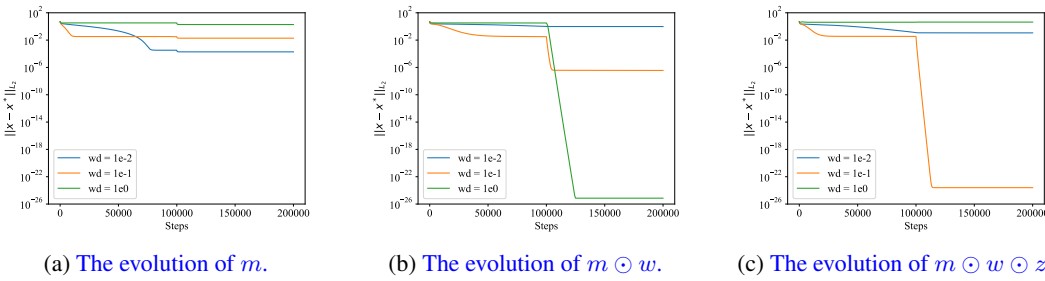

(a) The evolution of $m$.  (b) The evolution of $m \odot w$.  (c) The evolution of $m \odot w \odot z$.

Figure 5: Illustration of the effect of weight decay with higher order reparameterizations on generalization performance.

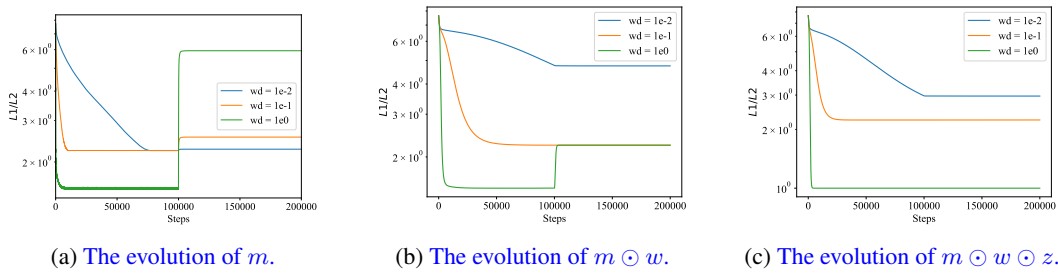

(a) The evolution of $m$.  (b) The evolution of $m \odot w$.  (c) The evolution of $m \odot w \odot z$.

Figure 6: The ratio between the $L_1$ and $L_2$ for diagonal linear networks.

**The reparameterization** $\log(u) - \log(v)$  In this paragraph, we consider another reparameterization. In the main text, we have seen that the regularization changed the type of bias from $L_2$ to $L_1$. We now consider a reparameterization with explicit regularization that leads to the opposite type of bias change. The reparameterization is $g(w) = \log(u) - \log(v)$. The regularization found in Corollary 3.1 is $h(w) = \sum_{i=1}^{n} \log(u_i) + \log(v_i)$. Then for $u, v > 1$ we can apply Theorem 3.1.

We now give the resulting time-dependent Legendre function. The time-dependent Legendre function is

$$R_a(x) = \frac{1}{4} \sum_{i=1}^{n} \left( u_{0,i}^2 - 2a \right) \log \left( e^{-2x_i} + 1 \right) + \left( v_{0,i}^2 - 2a \right) \log \left( e^{2x_i} + 1 \right) \ \forall a < \frac{1}{2} \min\{u_{0,i}^2, v_{0,i}^2\}.$$

The global minimum is centered at $\nabla_x R_a = 0$ and is given by $\log \left( \sqrt{u_0^2 - 2a} \right) - \log \left( \sqrt{v_0^2 - 2a} \right)$. Thus a shift occurs when $a$ changes, illustrating the positional bias. Moreover, to illustrate the type change, consider the balanced initialization $u_0 = v_0 = \beta I$, the Legendre function is then given by

$$R_a(x) = \frac{1}{4} \left( \beta^2 - 2a \right) \sum_{i=1}^{n} \log \left( 2 \cosh(x_i) \right)$$

which resembles the log-cosh loss function with vertical rescaling. The rescaling changes the type of bias from $L_1 \rightarrow L_2$. The type here is $L_2$ close to the origin and $L_1$ further away from zero. Due to the scaling, it becomes closer and closer to $L_2$. This is illustrated in Figure.7. Furthermore, we will show in experiments that the type change is crucial for generalization.

**Experiment** $\log(u) - \log(v)$.  In this context, we reparameterize the sparse code as $g(w) = \log(u) - \log(v) \in \mathbb{R}^n$ and replace the regularization as discussed. We initialize the parameters as $u_0 = 1/(\beta(1 + e^{-x}))$ and $v_0 = 1/(\beta(1 + e^x))$, where $\beta = 1$ and $x = 0.1$. Note, the initialization is different for stability reasons. We explore various values for $\alpha \in \{0.0001, 0.001, 0.01, 0.1, 0.0, 1.0\}$. During the training process, we track two key metrics: the reconstruction Mean Squared Error (MSE) and the nuclear norm of the sparse code, defined as $g(w) = \log u - \log v$. The results are illustrated in Figure.8. We observe that higher regularization leads to a faster increase in the nuclear norm, which confirms the movement to $L_2$ regularization. This leads to a construction error.

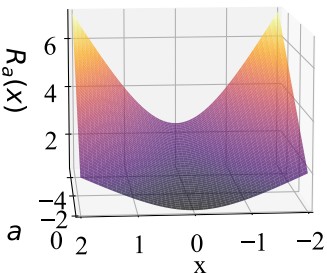

Figure 7: From $L_1$ to $L_2$ implicit bias, with $a = -\int_0^t \alpha_s ds$.

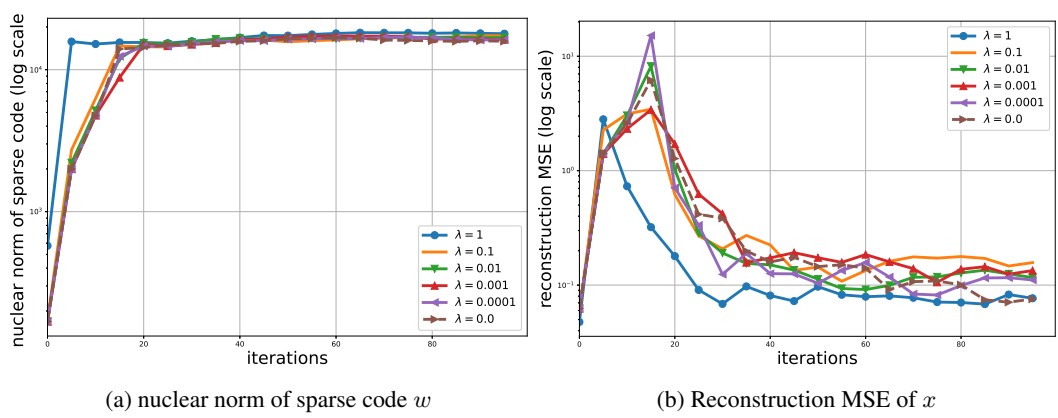

(a) nuclear norm of sparse code $w$

(b) Reconstruction MSE of $x$

Figure 8: Results for sparse coding reparameterisation $g(w) = \log u - \log v$

## C   HYPERPARAMETERS AND ADDITIONAL FIGURES

We present the experimental details in this section and an additional plot of the validation error. Moreover, we present the evolution of the time depenent Legendre functions corresponding to $m \odot w$ and $u^4 - v^4$ in Figures.9a and 9b .

For sparse coding we have used a learning rate $\eta = 0.001/Lip(D)$ where $Lip(D)$ denotes the resulting Lipschitz constant of the optimization problem depending on the dictionary $D$. In addition, we set the number of features $n = 50$ and run for 100 iterations. In the case of attention, we used the optimizer AdamW with learning rate $1e - 3$ and CosineAnnealingWarmRestarts. Finally, for LoRA we use SGD with momentum (0.9), constant learning rate $2e - 4$, LoRA rank 8, alpha 32 and no drop-out.

**Learning rate schedule and type of bias change**   We further study the effect of learning rate scheduler. Specifically, we run pre-trained ViT-tiny on ImageNet classification fine-tuning task. We set the learning rate to $1e - 4$ with AdamW optimisers. We also vary the weight decay in the range $[0.001, 0.003, 0.005, 0.007, 0.01]$. Moreover, for each of the settings, we train two comparison experiments, one without a learning rate scheduler, and one with the popular CosineAnnealingWarm-Restarts. We use the cumulative sum of the learning rate schedule as the x-axis.  The results are shown in Figure.10. Furthermore, results with SGD optimizer are included in Figure.11. We observe in both figures that the validation accuracy increases for the decaying schedule in comparison to the constant schedule. Moreover, we again observe a decaying ratio, for stronger weight decay the ratio decreases more. We observe that the modulation of the ratio is very similar, especially for SGD. This indicates that the implicit and explicit regularization is the dominating contributing factor to controlling the ratio, instead of the learning rate. Note that the AdamW optimizer can have additional effects that also contribute to changing the ratio.

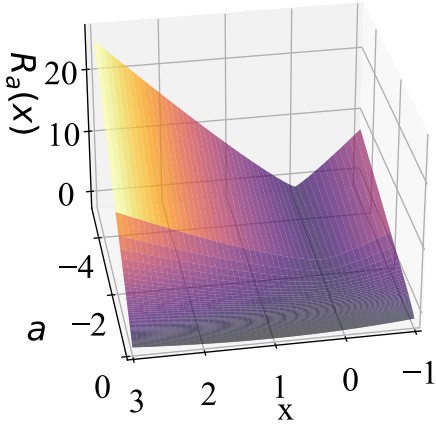
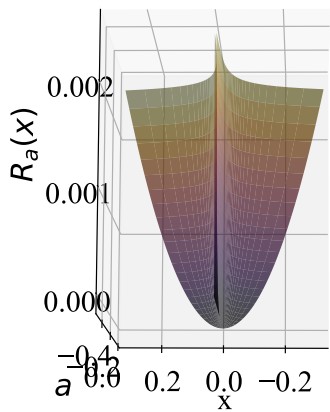

(a) The evolution of $R_a$ associated with $m \odot w$ initialized at $x_0 = 1$.

(b) The evolution of the approximated $R_a$ associated with $u^4 - v^4$ initialized at $x_0 = 0$.

Figure 9: Illustrations of the 3 effects of explicit regularization on the time-dependent Legendre function. In both figures $a = - \int_0^t \alpha_s ds$. The positional bias is illustrated in Fig 9a and the range shrinking is illustrated in Fig 9b. Both figures illustrate the change in type of bias from $L_2$ to $L_1$.

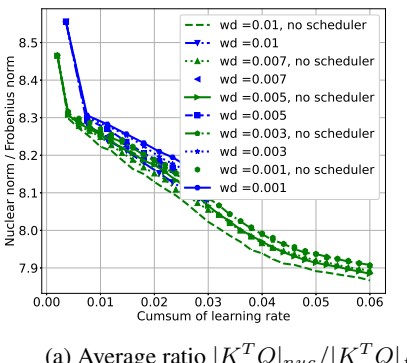
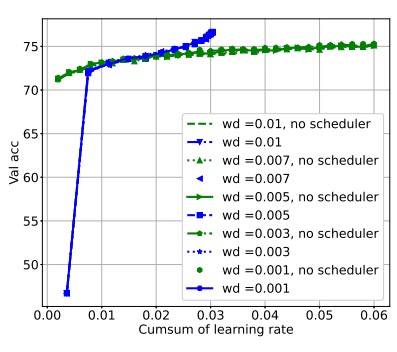

(a) Average ratio $|K^T Q|_{nuc}/|K^T Q|_{frob}$

(b) Validation accuracy

Figure 10: Results for ViT-tiny fine-tuning task with AdamW optimiser

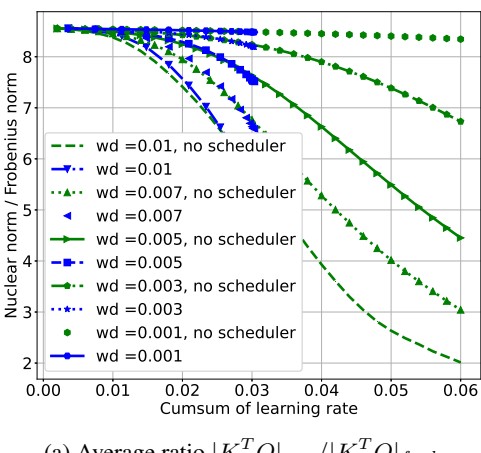
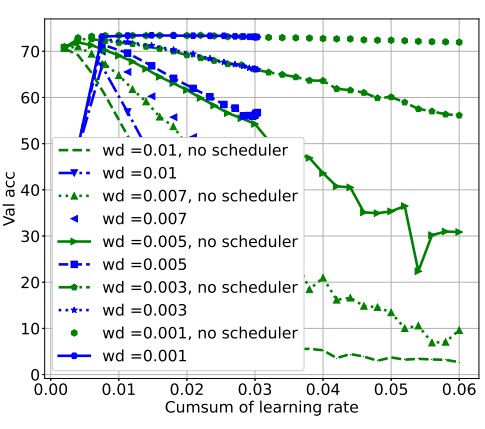

(a) Average ratio $|K^T Q|_{nuc}/|K^T Q|_{frob}$

(b) Validation accuracy

Figure 11: Results for ViT-tiny fine-tuning task with SGD optimiser

