# OpenReview forum: "Time-Dependent Mirror Flows and Where to Find Them"
_ICLR.cc/2025/Conference — Submitted to ICLR 2025_

### Official Review · Reviewer_fdqJ · 2024-10-31

**Soundness:** 3
**Presentation:** 2
**Contribution:** 2
**Rating:** 3
**Confidence:** 4

**Summary:**

Prior work has shown that, under some conditions, the optimization dynamics of overparameterized models can be formulated as a mirror flow (Li et al. 2022). The current work extends that framework to explicitly regularized losses. In particular, it establishes a sufficient condition under which gradient flow over the explicitly regularized objective can be described as a (possibly time-varying) mirror flow. This enables proving convergence to a minimizer when the loss function is convex. Then, for several types of overparameterizations, explicitly writes the potential of the mirror flow induced by certain explicit regularizers, and discusses implications to the resulting implicit bias.

Experiments support the theoretical results.

**Strengths:**

Explicit regularizers, such as $L_2$ regularization, are commonly applied in practice. Hence, understanding how they modify the implicit bias of optimization for a given model is an important question in the theory of deep learning. This paper makes progress towards addressing this question. While the logarithmic parameterization $\log (u) - \log (v)$ is quite unorthodox (it is not clear where one may encounter it), the $m \odot u$ and $u^{2n} - v^{2n}$ parameterizations have been widely studied in the past in the context of linear diagonal networks, and so may be of interest to the community.

**Weaknesses:**

1. Perhaps the main weakness of the paper is a rather severe lack of clarity, which in turn makes it difficult to assess the implications and significance of the results. As detailed below, some definitions are missing, notation is often used without proper introduction, some theorem statements are vague, terms such as "positional bias" are used without any intuitive or formal definition, and the implications of results are also not sufficiently discussed.

2. The significance of the results may be limited due to the parameterizations that they support. I find that the discussion around which parameterizations and regularizations are captured by the proposed framework and which are not can be strengthened. In particular, is there hope of using the framework for analyzing more practical models?

3. The fact that $L_2$ regularization in matrix factorization induces a bias towards low nuclear norm for the matrix product has been shown in prior work (see, e.g., [1]). The significance and novelty of the empirical observation for attention layers and LoRA is therefore limited, in my opinion (a discussion on the relation between $L_2$ regularization and nuclear norm in attention layers also appears in [2]).

Overall, while the paper has potential to further our understanding of how explicit regularization affects the implicit bias of optimization, at its current form I do not believe it is suitable for publication. I would recommend the authors to thoroughly revise the paper and improve its clarity, for which I hope the comments below can be useful. Furthermore, the significance of the results can be strengthened by further mapping out which parameterizations and regularizations fall under the proposed framework and which do not. Currently, the most practical models captured by the framework are matrix factorizations, which appear as part of different factorized layers. However, as mentioned above, the fact that an $L_2$ regularization translates to nuclear norm minimization for matrix factorization is already known (e.g., [1]) and does not necessitate the more complex machinery.

Detailed remarks regarding clarity:
- In Definition 3.1, it is not specified what the definition of a Legendre function is. I assume the intention was to use the same definition from Li et al. 2022. I recommend either stating the definition explicitly or at least referring to an existing definition in the literature (e.g. that from Li et al. 2022).

- In line 223, it is stated that $Q$ is defined in Theorem 3.1, however, it is not clear what the precise definition of $Q$ is. Seems that it is implicitly defined by $y = Q(x, a_t)$? Though since $y$ is actually $h(w)$ and $x$ is actually $g(w)$ there is no dependence on $a_t$, which leads me to believe the definition of $Q$ is not $Q(x, a_t) = h(w)$ as one may currently infer. Please explicitly define $Q$.

- In Definition 3.2, Bregman functions are not defined. I would recommend either explicitly defining this concept or at least referring to a specific place where they are defined. I assume the intended definition is Definition A.6 from Li et al. 2022.

- It can be useful to give examples for contracting parameterized Bregman functions around Definition 3.2 to help the reader understand this concept.

- In line 242 it is stated “assume that $\nabla f$ is locally Lipschitz and $argmin \\{ f(x) : x \in dom R_{a_\infty} \\}$”. It is not clear what the second part of the assumption is, that the argmin exists?

- In line 267 it is stated that the result for separable parameterizations encompasses “all previous settings”. Which previous settings does this refer to?

- In Corollary 3.1, what do $g_{i, j}$ and $w_{i, j}$ stand for? Is it assumed here that $g(w)$ and $w$ are matrices and that $i,j$ indexes an entry in them? In that case, what are $g_i$ and $w_i$?

- In Corollary 3.1, I find the statement “To apply Theorem 3.1 we need that…” to be too vague. Is the result that the conditions of Theorem 3.1 hold if and only if this holds? Or is it just a sufficient condition? Either way it is worth explicitly stating what the conditions are. An analogous comment applies to Corollary 3.2.

- In line 345, the regularization is defined as $y = m^2 + w^2$, where $m$ and $w$ are vectors, but isn’t the regularization $y = h(w)$ supposed to be a scalar?

- In line 361, it is not clear what “positional bias” means. This term is used in several places in the paper, but nowhere is it defined even on an intuitive level. What does “positional bias” refer to?

- The paragraph starting at line 364 claims to consider an attention layer. I find this to be quite misleading as the effect of the remaining components in an attention layer (value matrix and softmax) are not considered.

- In line 366, what are the $A_j$ matrices in this case? Are they not determined by the parameterization, in which case whether or not the condition holds is determined rather than something that you can/need to assume? It is also not clear where the nuclear norm comes from in Theorem 3.3.

- In line 411 it is stated that “The shift is centered at…”. What shift does this refer to? It is not clear what is being shifted and where.

- In line 418 it is stated that the rescaling changes the implicit bias from $L_1$ to $L_2$ regularization. It is worth clarifying why this is the case.

Additional (more minor comments):
- Typo in line 32: an unnecessary apostrophe.

- I find Figure 1 to be too vague (related to the remarks on clarity above). In particular, aside from the $L_2$ to $L_1$ change in implicit bias, it is not clear what “positional bias” or “range shrinking” refers to. Would be best to either make it more understandable from the figure itself or from the caption.

- In line 73, Pesme et al. 2024 is cited for showing that gradient descent converges to the solution with lowest $L_1$ distance from initialization for overparameterized linear regression. I believe this is not the correct citation, as Pesme et al. 2024 consider classification problems. Also, this claim is not true for all overparameterized linear models, rather for some parameterizations (e.g. linear diagonal networks) and under certain technical conditions (e.g. small initializations).


[1] Dai, Z., Karzand, M., & Srebro, N. Representation costs of linear neural networks: Analysis and design. NeurIPS 2021.

[2] Khodak, M., Tenenholtz, N., Mackey, L., & Fusi, N. Initialization and regularization of factorized neural layers. ICLR 2021.

**Questions:**

--

---

> ### Author Response · Authors · 2024-11-25
>
> We would like to express our gratitude for your time and efforts in providing valuable comments and detailed feedback on our manuscript. We have carefully reviewed the comments and have made corresponding revisions to our manuscript, in particular, to increase its clarity. Please find a detailed point-by-point response below. In case of any open questions or concerns, we would be happy to discuss them.
>
> **1. Resolved clarity issues:**
>
> 1.  Definition Legendre function:
>
> We have added a reference to Definition 3.1 (Li et al. 2022) when the Legendre function is introduced.
>
> 2. Discrepancy about $Q$:
>
> We define $Q$ as the inverse of the map $y \rightarrow \partial_y R(x,y)$. The dependence on $a_t$ comes from the (standard) mirror flow equation. Namely, we have $ \partial_y R(x_t,y_t) = a_t$. We have updated the statement to make this clear.
>
> 3. Definition Bregman function:
>
> We have added a reference to Definition A.6 (Li et al. 2022) where the Legendre function is introduced.
>
> 4. Contracting example:
>
> We have added an example when the definition is given. The example is $R_a(x) = (x-a)^2$, where it is contracting on the set $(-\infty, 0]$.
>
> 5. Existence of minimizer:
>
> We have reformulated it so that the set of minimizers must be non-empty.
>
> 6. Previous settings.
>
> We have added explicit references to previous work that uses the reparameterizations.
>
> 7. Definitions of $g_i$ and $w_i$:
>
> We have specified that $g_{i,j}: \mathbb{R} \rightarrow{R}$ and $h_{i,j} : \mathbb{R} \rightarrow \mathbb{R}$. This makes $g_i : \mathbb{R}^{m_i} \rightarrow \mathbb{R}$ and $w_i \in \mathbb{R}^{m_i}$.
>
> 8. Clarity of Corollaries 3.1 and 3.2:
>
> Both corollaries have been made precise. Theorem 3.1 applies if and only if the proposed condition holds. Furthermore, we have moved Corollary 3.2 to Appendix B.1 to improve the flow of the manuscript. We have also expanded the discussion on the limitations of the proposed framework there.
>
> 9. Notation of regularization.
>
> We agree that the current notation was not rigorous. We have updated all notations of regularization. For example, $m^2 + w^2$ is replaced with $||m||^2_{L_2} + ||w||^2_{L_2}$.
>
> 10. The 3 effects:
>
> In the revised manuscript, we define 3 main effects how explicit regularization impacts the implicit bias and described their interplay in the introduction section as follows:
>
> 1. Type of bias: Explicit regularization changes the shape of the Legendre function (and thus the nature of the implicit regularization). For example, the shape of the Legendre function changes from an $L_2$ norm to an $L_1$ norm. Especially if our goal is sparsification to leverage the implicit $L_1$ regularization, starting with $L_2$ regularization and thus a slow sparsification, can be critical in the context of deep learning, where training overparameterized models is known to boost performance.
>
> 2. Positional bias: The explicit regularization shifts the global minimum of the Legendre function. Usually, we expect the minimum to be zero (e.g. if we want to sparsify a model in case of $L_1$ type of bias). Yet, in the standard mirror flow Legendre function, the global minimum corresponds to the network's initialisation. During training, the explicit regularization can move the minimum closer to zero. Only when we move the minimum to 0, we would actually promote sparsity in case of implicit $L_1$ regularization, for instance. For that reason, it is important to ensure that the positional bias vanishes during training. We show that this can be achieved by explicit regularization.
>
> 3. Range shrinking: In addition to the positional bias, the explicit regularization also shrinks the range of the attainable values of the Legendre function. For example, the $L_1$ norm of the network parameters becomes fixed during training. This can be a problem if it hinders further training.
>
> Moreover, we have made plots for the analyzed time-dependent Legendre functions, see Figure 7 and 9.
>
>
> 11. Transformer mechanism:
>
> We have added a disclaimer in the text that there are more parts to consider in the case of attention. The analysis in (Sheen et al. (2024)) assumes for example that the value matrix is not trainable. The softmax operation can be seen as part of the function $f$. Therefore, in case of a non-trainable value matrix, Theorem 3.1, characterizing the implicit bias, applies.
>
> Sheen, H., Chen, S., Wang, T., & Zhou, H.H. (2024). Implicit Regularization of Gradient Flow on One-Layer Softmax Attention. ArXiv, abs/2403.08699.

---

> > ### Author Response · Authors · 2024-11-25
> >
> > 12. The choice of $A_i$:
> >
> > Indeed $A_i$ have to be chosen. We now have made this clear in the text, and discuss particular choices such as diagonal matrices and the broader class of matrices satisfying the alignment property. Based on the diagonal matrices, we illustrate that the type of bias for the eigenvalues of $K^TQ$ changes from $L_2$ to $L_1$. This implies for the general matrix that the type of bias changes from the Frobenius norm to the nuclear norm.
> >
> > 13. The shift:
> >
> > The shift refers to the time-dependent Legendre function’s global minimum. This illustrates the effect of the positional bias, which we now explain in the introduction. Moreover, we have moved the discussion on the reparameterization $log(u) - log(v)$ to the appendix to improve the flow of the paper.
> >
> > 14. Scaling effect:
> >
> > To illustrate the effect of the scaling on the type of bias we have added Figure 7, where we plot the time-dependent Legendre function.
> >
> > Minor comments:
> > We have addressed these comments in the manuscript.
> >
> > **2. Limitations of the framework**
> >
> > We have relocated the limitations of the proposed framework to Appendix B1 and expanded on these limitations by analyzing the parameterization $g(w) = \Pi_{i = 1}^k w_i$ with weight decay regularization $h(w) = \sum_{i=1}^k w_i^2$. In this analysis, we demonstrate that it lies outside the scope of the current framework, rendering Theorem 3.1 inapplicable. Nevertheless, we can transfer our insights for $m\odot w$ and $u^{2k} - v^{2k}$ to predict the implicit bias. We observe in a diagonal linear network experiment that both the type of bias and range shrinking effect are present. Specifically, when regularization is turned off, the sparse ground truth is recovered if enough regularization was applied beforehand, as observed with the $m \odot w$ parameterization. However, excessive regularization causes the range shrinking effect to dominate and prevents reaching the ground truth, as similarly observed with  $u^{2k} - v^{2k}$. Note that the solution $x$ is distinct from both the ground truth and $0$, which is significant as the dynamics might get stuck at zero.
> >
> > **3. Novelty of insight into quadratic parameterizations**
> >
> > The cited papers primarily focus on the induced model capacity of regularization with reparameterization and the difference between weight decay and Frobenius regularization. However, they do not analyze the impact of explicit regularization onto the implicit bias, nor do they systematically derive the nature of the implicit bias, which as we show, changes from an $L_2$ to $L_1$ or Frobenius norm to nuclear norm during training.
> >
> > In addition to our theoretical extensions, we gain novel and practical insights concerning **quadratic reparameterizations:**
> > 1. The type of bias changes from an $L_2$ to $L_1$ Legendre function, as observed in our experiments. This insight can only be attained with our dynamic view. The time-dependent Legendre function changes during training, which changes the geometry of the training dynamics as described in Equation (7). This could have different implications, e.g., on the effect of early stopping, or explaining scaling laws that relate the amount of overparameterization to optimal training times (for best generalization), or guide simply dynamic tuning schedules for weight decay. Note that the type of bias is conceptually different from model capacity and is induced by a time-dependent Legendre function. We do not observe a fixed model capacity (with respect to either $L_2$ or $L_1$), as it changes from one to the other during training.
> > 2. The effect of the regularization is determined by the time-dependent Legendre function. This allows us to turn off weight decay at the end of the training while still keeping the effect of the regularization. We have extended the **LoRA** experiment to illustrate this point more clearly. Now we turn off the weight decay after $200$ iterations for $2$ of the $4$ configurations. We observe that the ratio of the nuclear norm and Frobenius norm still decreases despite switched-off weight decay, effectively storing the regularization. This allows us to train longer with a desired norm ratio but unconstrained by the explicit regularization. This insight can be used to design better regularization schedules. For example, turning off the regularization at a desired ratio. In our LoRA experiments, this approach attains the models of lowest rank without drop in performance.

---

> > > ### Comment · Reviewer_fdqJ · 2024-11-25
> > >
> > > Thank you for the detailed response, I have read it and the other reviews carefully. As stated in my review, the original manuscript suffered from major issues in clarity and also in terms of the significance of some of the contributions. I appreciate the effort in updating the paper during the discussion period to address these concerns. It seems that the updated version has improved upon the initial submission, however, I believe the extent of changes essentially necessitates another full review of the paper, and is not suitable for the author-reviewer discussion period. Unfortunately, I therefore cannot recommend acceptance of the paper.
> > >
> > > Regarding the novelty of identifying that $L_2$ regularization in matrix factorization (equivalently, linear neural networks) leads to a bias towards low nuclear norm. I agree that technically the result in the cited work is not identical to that provided in this paper. Though, it is worth noting that it does imply that using an $L_2$ regularization means you learn a solution which has minimal nuclear norm, out of all possible matrices that attain the same loss. In that sense it is not true that the insight of $L_2$ explicit regularization being converted to a nuclear norm regularization can only be attained via the dynamic view proposed in this work. With that said, the other potential implications regarding early stopping or explaining scaling laws are not explained by prior work. It may be worth emphasizing them.

---

> > > > ### Author Response · Authors · 2024-11-26
> > > >
> > > > Thank you for your response and for thoroughly reviewing the other comments.
> > > >
> > > > We have enhanced the clarity of our manuscript based on your valuable suggestions and have added further explanations to make the material better accessible for non-experts. We are happy that our main storyline was accessible to you and we are impressed by your deep understanding of the subject and our work. Please note that the main content of the paper remains unchanged. All revisions have been detailed in the accompanying comments, and as such, a complete review of the manuscript is not required.
> > > >
> > > > Our novel insight in comparison to the cited paper is that we find that the Legendre function is time-dependent and evolves from $L_2$​ to $L_1$​ regularization. This implies that the implicit bias is not $L_1$. It depends on the strength of the weight decay and training time what regularization we encounter. It can be $L_2$ or $L_1$ or something in between. For that reason, we would also receive different results if we were to train a model (with original parameters $x$) with constant $L_1$ regularization.

---

### Official Review · Reviewer_LGEW · 2024-11-01

**Soundness:** 2
**Presentation:** 1
**Contribution:** 2
**Rating:** 3
**Confidence:** 2

**Summary:**

The paper studies mirror flows, with the goal of identifying how explicit regularization and the inducing parameterization affect implicit regularization.
For certain parameterizations and regularizations, it characterizes their impact.

**Strengths:**

- **S1: Novelty and generality** Making the interplay of explicit and implicit
  regularization concrete is certainly a relevant goal, as it provides the
  possibility to modulate the explicit regularization with time to achieve a
  specific implicit regularization. The mirror flow framework, in which the
  paper expands on this goal, provides a general framework, and the presented
  results seem general.

**Weaknesses:**

I found the paper extremely challenging to follow. This may be because I am only
partially familiar with mirror flows (reflected by my confidence), but I also
believe the presentation is at many points not self-consistent, lacking clarity,
and missing interpretation and relation to practical contexts. In the following,
I am pointing out some aspects that prevented me from comprehending and
appreciating the paper's contribution:

- **W1 Self-consistency and clarity:** The introduction to mirror flows is
  confusing.

  Specifically, I struggled to keep up with relating the presented math in
  Sections 3+4 to that in the introduction: the primal objective $f(x)$,
  (Equation 4), primal explicit regularization $h(x)$ (Equation 4), dual
  objective $f(g(w))$, and dual explicit regularization $h(w)$ (Equation 5). I
  think the paper would be much clearer if Sections 3+4 used notation that made
  the connections to these objects obvious. Sometimes, it is also hard to follow
  along because objects do not have names, for instance there is a function $R$
  that is always referred to as 'the function $R$'; is there a more meaningful
  name, e.g. is it related to the distance-generating function in mirror
  descent? Same with the 'Bregman potential', that is mentioned in the abstract,
  but does not seem to show up in the main text. I also found the introduction
  to mirror flows in A.1 not very helpful as it mainly consists of an
  enumeration of definitions.

  Another problem I had in keeping up with the presentation is that some of the
  steps were not clearly motivated or outlined. Therefore, I could only
  acknowledge the existence of some results, specifically the examples on
  different parameterizations in Examples 3.1+3.2, as well as their paragraphs
  in Sections 4+5, but not understand how they contribute to the bigger picture.

  Lastly, the authors introduce the concepts of positional bias and range
  shrinking in the introduction, but never explain what exactly they mean.

- **W2 Interpretation and relation to practical contexts** I could not follow
  certain interpretations, explanations, and motivations, e.g. why is it
  interesting to study the setup in Corollary 3.2? Are the parameterizations
  presented in the examples relevant in practise? How does the paragraph below
  Equation 8 relate to its math? What is the theory's prediction for LoRA in
  Section 5, and is it reflected by the empirical results? I think these are
  currently just not explained clearly enough, making it hard to assess the
  paper.

**Questions:**

Please see W1 and W2.

I am willing to re-assess my score if the authors find ways to clarify the presentation and motivation.

---

> ### Author Response · Authors · 2024-11-25
>
> We would like to express our gratitude for your time and efforts in providing valuable comments and detailed feedback on our manuscript. We have carefully reviewed the comments and have made corresponding revisions to our manuscript, in particular, to increase its clarity. Please find a detailed point-by-point response below. In case of any open questions or concerns, we would be happy to discuss them.
>
>
> **W1. Self-consistency and clarity:**
>
> We have clarified the connections between the primal and dual objectives by discussing the dual gradient flow associated with Eq. (1), which leads directly to the primal mirror flow described in Eq. (4). Additionally, we present the dual gradient flow corresponding to Eq. (3) prior to introducing Theorem 3.1, where we derive the associated primal time-dependent mirror flow. To ensure clarity, we have added subscripts to all gradients and Hessians indicating if it is with respect to the primal or dual variable. Moreover, we now explicitly and consistently refer to $g$ as the reparameterization and $h$ as the explicit regularization.
>
> The implicit regularizer $R$ can indeed be interpreted as a distance-generating function, akin to those used in mirror descent. In the revised manuscript, we now refer to the function $R$ as the Legendre function or time-dependent Legendre function. For additional context, we also mention the link to mirror descent. Note, in the specific case for convergence and quadratic parameterizations, $R$ becomes a Bregman function. For precise definitions, we refer to (Li et al. (2022)), where both the definition of a Legendre and Bregman function can be found in Definitions A.6 and 3.2. Finally, we agree that the term “potential” can be confusing and have replaced it by “function”.
>
> We refer to previous works to put the considered parametrizations in greater context which we mention in W2.
>
> In the revised manuscript, we define 3 main effects how explicit regularization impacts the implicit bias and described their interplay in the introduction section as follows:
>
> 1. Type of bias: Explicit regularization changes the shape of the Legendre function (and thus the nature of the implicit regularization). For example, the shape of the Legendre function changes from an $L_2$ norm to an $L_1$ norm. Especially if our goal is sparsification to leverage the implicit $L_1$ regularization, starting with $L_2$ regularization and thus a slow sparsification, can be critical in the context of deep learning, where training overparameterized models is known to boost performance.
>
> 2. Positional bias: The explicit regularization shifts the global minimum of the Legendre function. Usually, we expect the minimum to be zero (e.g. if we want to sparsify a model in case of $L_1$ type of bias). Yet, in the standard mirror flow Legendre function, the global minimum corresponds to the network's initialisation. During training, the explicit regularization can move the minimum closer to zero. Only when we move the minimum to 0, we would actually promote sparsity in case of implicit $L_1$ regularization, for instance. For that reason, it is important to ensure that the positional bias vanishes during training. We show that this can be achieved by explicit regularization.
>
> 3. Range shrinking: In addition to the positional bias, the explicit regularization also shrinks the range of the attainable values of the Legendre function. For example, the $L_1$ norm of the network parameters becomes fixed during training. This can be a problem if it hinders further training.
>
> Moreover, we have made plots for the analyzed time-dependent Legendre functions, see Figures 7 and 9.
>
>
> Li, Z., Wang, T., Lee, J.D., & Arora, S. (2022). Implicit Bias of Gradient Descent on Reparametrized Models: On Equivalence to Mirror Descent. ArXiv, abs/2207.04036.

---

> > ### Author Response · Authors · 2024-11-25
> >
> > **W2: Interpretation and relation to practical contexts:**
> >
> > The main practical implication of our theoretical insights is that we can change and control the implicit bias by explicit regularization, even dynamically during training. Specifically, we use our findings to derive guiding principles for sparse coding, attention, and LoRA finetuning. For instance, we gain insights into how weight decay promotes effectively low rank solutions. While attention requires sufficiently strong regularization for $L_1$ to dominate late in training, LoRA finetuning finds low rank solutions if weight decay is turned off in later training rounds. More details on the exposition follow below.
> >
> > **Different parameterizations:** The goal of Corollary 3.2 is to explore the limitations of the proposed framework. We have moved Corollary 3.2 to the appendix and have provided an additional class of parameterizations. The additional class is described by the reparameterization $g(w) = \Pi_{i = 1}^k w_i$ and $h(w) = \sum_{i = 1}^k w_i^2$, with $k >2$. We show that this class does not satisfy the sufficient condition to apply Theorem 3.1, which characterizes the time-dependent mirror flow. Nevertheless, we can make predictions based on the developed framework, which we demonstrate in the context of diagonal linear network experiments.  We observe both the type of bias and range shrinking effect for the reparameterization with $k = 3$. The type of bias prediction follows from the analyzed reparameterization $m \odot w$ and the range shrinking follows from the reparameterization $u^{2k} - v^{2k}$.
> > We have included references to previous work where specific reparameterizations were studied to highlight the broader relevance of our framework. For instance, reparameterizations such as $m \odot w$ (Pesme et al. 2021) and $u^{2k} -v^{2k}$ (Woodworth et al. 2020)  have been used to gain insights into the training dynamics of neural networks, specifically, the effect of stochasticity and overparameterization. They have also been exploited for sparsification (Jacobs & Burkholz 2024).
> >
> > Pesme, S., Pillaud-Vivien, L., & Flammarion, N. (2021). Implicit Bias of SGD for Diagonal Linear Networks: a Provable Benefit of Stochasticity. Neural Information Processing Systems.
> >
> > Woodworth, B.E., Gunasekar, S., Lee, J., Moroshko, E., Savarese, P.H., Golan, I., Soudry, D., & Srebro, N. (2019). Kernel and Rich Regimes in Overparametrized Models. ArXiv, abs/2002.09277.
> >
> > Jacobs, T., & Burkholz, R. (2024). Mask in the Mirror: Implicit Sparsification. ArXiv, abs/2408.09966.
> >
> > For the derivation of Eq. (8), we use Theorem 3.3. The main step is inverting the function $Q_a(\mu)$. We added an explanation after the equation.
> >
> > **Practically relevant insights:**
> > We gain two key novel insights of our intricate extension for **quadratic reparameterizations:**
> > 1. The type of bias changes from an $L_2$ to $L_1$ Legendre function, as observed in our experiments. This insight can only be attained with our dynamic view. The time-dependent Legendre function changes during training, which changes the geometry of the training dynamics as described in Equation (7). This could have different implications, e.g., on the effect of early stopping, or explaining scaling laws that relate the amount of overparameterization to optimal training times (for best generalization), or guide simply dynamic tuning schedules for weight decay. Note that the type of bias is conceptually different from model capacity and is induced by a time dependent Legendre function. We do not observe a fixed model capacity (with respect to either $L_2$ or $L_1$), as it changes from one to the other during training.
> > 2. The effect of the regularization is determined by the time-dependent Legendre function. This allows us to turn off weight decay at the end of the training while still keeping the effect of the regularization. We have extended the **LoRA** experiment to illustrate this point more clearly. Now we turn off the weight decay after $200$ iterations for $2$ of the $4$ configurations. We observe that the ratio of the nuclear norm and Frobenius norm still decreases despite switched-off weight decay, effectively storing the regularization. This allows us to train longer with a desired norm ratio but unconstrained by the explicit regularization. This insight can be used to design better regularization schedules. For example, turning off the regularization at a desired ratio. In our LoRA experiments, this approach attains the models of lowest rank without drop in performance.

---

> ### Comment · Reviewer_LGEW · 2024-11-25
>
> Thank you for your detailed response! I have no follow-up questions for now.

---

> > ### Author Response · Authors · 2024-12-03
> >
> > We would like to thank you for your valuable review. We believe that we have addressed all points of criticism but would be happy to address any open issues if there should remain any. Since the discussion period is approaching its end, we would highly appreciate your feedback.

---

### Official Review · Reviewer_ooDS · 2024-11-03

**Soundness:** 3
**Presentation:** 3
**Contribution:** 3
**Rating:** 6
**Confidence:** 4

**Summary:**

This paper show that explicit regularization modifies the behavior of implicit bias and provides a mechanism to control its strength. The primary theoretical contribution is the characterization of regularizations and parameterizations that induce a time-dependent Bregman potential, with a discussion of the implications of its temporal variation. This framework encompasses single-layer attention, and application to sparse coding.

**Strengths:**

1. This paper presents sufficient conditions for incorporating explicit regularizations into the mirror flow framework and characterizes their effects by analyzing three main aspects of implicit bias: shifts in positional bias, the type of bias introduced, and the reduction in the range of values.

2. This paper proposes a systematic method for identifying these regularizations and establishes a general convergence result within this framework.

3. This paper emphasizes the impact of regularization on implicit bias, demonstrating these effects in experiments such as sparse coding, transformer attention mechanisms, and LoRA fine-tuning in large language models.

4. This paper reveals that weight decay modulates the degree of sparsification brought about by quadratic parameterizations, including those in attention mechanisms and LoRA adjustments.

**Weaknesses:**

1.However, their interplay remains poorly understood. This inference lack support.

2. More experiments on real world datasets are needed to verify the validity of their method.

3. This paper lacks the motivation about the behavior of implicit bias

**Questions:**

See weaknesses.

---

> ### Author Response · Authors · 2024-11-25
>
> We would like to express our gratitude for your time and efforts in providing valuable comments on our manuscript. We have carefully reviewed the comments and have made corresponding revisions to our manuscript in response to your insightful feedback. Please find a detailed point-by-point response below. In case of any open questions or concerns, we would be happy to discuss them.
>
> **1. Three main effects of explicit regularization on implicit bias:**
>
> In the revised manuscript, we define 3 main effects how explicit regularization impacts the implicit bias and described their interplay in the introduction section as follows:
>
> 1. Type of bias: Explicit regularization changes the shape of the Legendre function (and thus the nature of the implicit regularization). For example, the shape of the Legendre function changes from an $L_2$ norm to an $L_1$ norm. Especially if our goal is sparsification to leverage the implicit $L_1$ regularization, starting with $L_2$ regularization and thus a slow sparsification, can be critical in the context of deep learning, where training overparameterized models is known to boost performance.
>
> 2. Positional bias: The explicit regularization shifts the global minimum of the Legendre function. Usually, we expect the minimum to be zero (e.g. if we want to sparsify a model in case of $L1$ type of bias). Yet, in the standard mirror flow Legendre function, the global minimum corresponds to the network's initialisation. During training, the explicit regularization can move the minimum closer to zero. Only when we move the minimum to 0, we would actually promote sparsity in case of implicit $L1$ regularization, for instance. For that reason, it is important to ensure that the positional bias vanishes during training. We show that this can be achieved by explicit regularization.
>
> 3. Range shrinking: In addition to the positional bias, the explicit regularization also shrinks the range of the attainable values of the Legendre function. For example, the $L_1$ norm of the network parameters becomes fixed during training. This can be a problem if it hinders further training.
>
> We have included additional experiments in Appendix B1 to demonstrate the three explained effects. For example, Figure 4b in the revised paper shows that, with moderate regularization, there is a clear type-of-bias change from $L_2$ norm to $L_1$ norm. In contrast, higher levels of regularization do not exhibit the same behavior, which we attribute to the range-shrinking effect. In consequence, for higher levels of regularization we have found an additional contributor that reduces the trainability of the neural network. Moreover, we have made plots for the analyzed time-dependent Legendre functions, see Figures 7 and 9.
>
> **2. More experiments on real world data:**
>
> We have extended the LoRA experiment. In this revised setup, weight decay is turned off after $200$ iterations for $2$ of the $4$ configurations. Interestingly, we observe that the ratio between the nuclear norm and Frobenius norm continues to decrease, effectively storing the regularization effect within the time-dependent Legendre function. This behavior allows us to train longer with a desired ratio unconstrained by the explicit regularization. Therefore, this insight can be used to design better regularization schedules (e.g., turning off regularization once a desired ratio is achieved).
>
> Furthermore, we have included an additional experiment on ImageNet with a transformer model in Appendix C. We finetune a pretrained model with varying weight decay and two learning rate schedules. To fairly compare the different learning rate schedules, we use the cumulative sum of the learning rate schedules as the x-axis of the plots. The results show that ratios decay as predicted by the theory, i.e., higher weight decay leads to a large decrease in the ratio. Moreover, we observe that for each weight decay configuration, both learning rate schedules lead to a similar decrease, especially for SGD.

---

> > ### Author Response · Authors · 2024-11-25
> >
> > **3. Motivation and significance of implicit bias framework:**
> >
> > The main goal of characterizing implicit bias is to understand the role of overparameterization in deep learning and how it contributes to its success. Commonly, the corresponding regularization is tied to the specifics of the parameterization, neural network architecture, and the optimizer. Yet, its strength is usually assumed to be fixed during training.
> >
> > Our framework extension changes this picture, as it aims to understand the effect of explicit regularization on the implicit bias (which is almost always applied in practice). This is significant, as the explicit regularization changes the nature and strength of the implicit bias of the studied architecture and gives us a way to control and therefore exploit it. This nature and strength can even change dynamically during training. Importantly, this dynamic change is critical to address issues with range shrinking and positional bias, and thus guarantee convergence.
> >
> > From a conceptual point of view, our results are of independent interest, as they extend the mirror flow framework to a time-dependent mirror flow framework.
> >
> > From a practical point of view, we gain insights into modern architectural characteristics such as attention and LoRA, and how weight decay induces sparsity. To boost their performance, we propose to switch-off weight decay in the last training rounds. (Note that this switch-off enables convergence of a minimizer of the original optimization objective $f$ according to Theorem 3.2.)
> >
> > We have highlighted these contributions more clearly in the introduction of our revised manuscript.
> >
> > **Practically relevant insights:**
> > We gain two key novel insights of our intricate extension for **quadratic reparameterizations:**
> > 1. The type of bias changes from an $L_2$ to $L_1$ Legendre function, as observed in our experiments. This insight can only be attained with our dynamic view. The time-dependent Legendre function changes during training, which changes the geometry of the training dynamics as described in Equation (7). This could have different implications, e.g., on the effect of early stopping, or explaining scaling laws that relate the amount of overparameterization to optimal training times (for best generalization), or guide simply dynamic tuning schedules for weight decay. Note that the type of bias is conceptually different from model capacity and is induced by a time dependent Legendre function. We do not observe a fixed model capacity (with respect to either $L_2$ or $L_1$), as it changes from one to the other during training.
> > 2. The effect of the regularization is determined by the time-dependent Legendre function. This allows us to turn off weight decay at the end of the training while still keeping the effect of the regularization. We have extended the **LoRA** experiment to illustrate this point more clearly. Now we turn off the weight decay after $200$ iterations for $2$ of the $4$ configurations. We observe that the ratio of the nuclear norm and Frobenius norm still decreases despite switched-off weight decay, effectively storing the regularization. This allows us to train longer with a desired norm ratio but unconstrained by the explicit regularization. This insight can be used to design better regularization schedules. For example, turning off the regularization at a desired ratio. In our LoRA experiments, this approach attains the models of lowest rank without drop in performance.

---

> > > ### Comment · Reviewer_ooDS · 2024-11-26
> > > **Official Comment by Reviewer ooDS**
> > >
> > > I maintain my scores after reading the authors' responses and other reviewers' comments.

---

### Meta-Review · Area_Chair_3LuN · 2024-12-19

**Metareview:**

In this work the authors consider the effect that explicit regularization has on the implicit regularization induced by optimization dynamics.  This is an interesting question, as adding explicit regularization clearly changes the optimization dynamics and by extension the implicit regularization, which is a worthwhile topic to understand in more detail.  Unfortunately, however, many of the reviewers found the clarity of the manuscript to be lacking to the point of being difficult to assess the significance of the contribution.  Beyond manuscript clarity there are also questions about whether the formulations and parameterizations studied will be of interest for practical problems.  For these reasons I am rejecting the paper but would encourage the authors to clarify their presentation along with the motivation/justification for the studied formulations and parameterizations for submission to a future conference.

**Additional Comments On Reviewer Discussion:**

The authors appear to have made substantial modifications to the paper in response to the initial reviews.  However, while this was appreciated by the reviewers, many of the edits were to technical definitions and statements/claims.  Given the extent of the changes the reviewers suggested that a full re-review of the paper would be required in a future venue.

---

### Decision · Program_Chairs · 2025-01-22

Reject